# Ideas and perspectives: Sea-Level Change, Anaerobic Methane Oxidation, and the Glacial-Interglacial Phosphorus Cycle

Bjorn Sundby [1,2], Pierre Anschutz [3], Pascal Lecroart [3], and Alfonso Mucci [2]

[1]ISMER, Université du Québec à Rimouski, Rimouski, QC, Canada H4C 3J9
[2]GEOTOP and Earth& Planetary Sciences, McGill University, Montreal, QC, Canada H3A 0E8
[3]Univ. Bordeaux, CNRS, EPOC, EPHE, UMR 5805 F-33615 Pessac, France

*Correspondence to*: Pierre Anschutz  (pierre.anschutz@u-bordeaux.fr)

**Abstract.** The oceanic phosphorus cycle describes how phosphorus moves through the ocean, accumulates with the sediments on the seafloor, and participates in biogeochemical reactions. We propose a new two-reservoir scenario of the glacial-interglacial phosphorus cycle. It relies on diagenesis in methane hydrate-bearing sediments to mobilize sedimentary phosphorus and transfer it to the oceanic reservoir during times when falling sea level lowers the hydrostatic pressure on the seafloor and destabilizes methane hydrates. The stock of solid phase phosphorus mobilizable by this process is of the same order of magnitude as the dissolved phosphate inventory of the current oceanic reservoir. The potential, additional flux of phosphate during the glacial period is of the same order of magnitude as pre-agricultural, riverine dissolved phosphate fluxes to the ocean. Throughout the cycle, primary production assimilates phosphorus and inorganic carbon into biomass which, upon settling and burial, returns phosphorus to the sedimentary reservoir. Primary production also lowers the partial pressure of $CO_2$ in the surface ocean, potentially drawing down $CO_2$ from the atmosphere. Concurrent with this slow 'biological pump', but operating in the opposite direction, a 'physical pump' brings metabolic $CO_2$-enriched waters from deep-ocean basins to the upper ocean. The two pumps compete, but the direction of the $CO_2$ flux at the air-sea interface depends on the nutrient content of the deep waters. Because of the transfer of reactive phosphorus to the sedimentary reservoir throughout a glaciation cycle, low phosphorus/ high $CO_2$ deep waters reign at the beginning of a deglaciation, resulting in rapid transfer of $CO_2$ to the atmosphere. The new scenario provides another element to the suite of processes that may have contributed to the rapid glacial-interglacial climate transitions documented in paleo-records.

## 1 Introduction

Each of the glacial cycles that characterize the Pleistocene lasted roughly 100 Kyr. During each cycle, about 90 Kyr were glacial (cold), allowing continental ice sheets to build up, and 10 Kyr were interglacial (warm), allowing the ice sheets to melt. The large volumes of water transferred from the ocean to the continents during the glacial part of the cycle and the return flow of water to the ocean during the interglacial caused the globally-averaged sea level to fall by more than 100 meters and then rise again (Siddal et al., 2003; Lambeck et al., 2014). Seasonality, caused by cyclic variations in Earth's orbit around the Sun (Milankovitch cycles), are the fundamental drivers of the glacial cycles, but orbital variations by

themselves cannot account for the rapid climate transitions documented in paleo-records. This implies that positive feedbacks within Earth's climate system must amplify the orbital drivers (Sigman and Boyle, 2000). Much research has focused on processes that might provide such feedbacks. Given that carbon dioxide ($CO_2$) is a greenhouse gas and since the ocean is the largest $CO_2$ reservoir on Earth (fifty times larger than the atmospheric reservoir), processes that influence the
exchange of $CO_2$ between the ocean and the atmosphere top the list of candidates.

Measurements on air samples trapped in ice cores have shown that the partial pressure of $CO_2$ ($pCO_2$) in the atmosphere during the glacial period of a cycle was substantially lower than during the intervening interglacial (Barnola et al., 1987; Petit et al., 1999). The atmospheric $pCO_2$ decreased slowly throughout the glacial part of the cycle, as did sea level. By comparison, both the rise of atmospheric $pCO_2$ and the sea-level rise at the end of the glaciation (termination) were
remarkably rapid, lasting 10 Kyr or less. The concomitant change in sea level and atmospheric $pCO_2$ through time suggests that the two phenomena are linked, but the nature of the link has remained elusive (Sigman and Boyle, 2000; Kohfeld et al., 2005; Peacock et al., 2006; Sigman et al., 2010; Kölling et al., 2019).

Another greenhouse gas, methane, is abundant in continental margin sediments where it occurs as a solid in the form of methane hydrate (clathrate) (Kvenvolden, 1993; Bohrmann and Torres, 2006; Ruppel and Kessler, 2017). Paull et al. (1991)
proposed a direct link between sea-level fall and global warming as follows: As the sea level falls, the pressure on the sediment column decreases, methane hydrates become destabilized, and methane gas is released into the pore water. They further proposed that methane released in this way could reach the atmosphere and initiate a warming event, but it is doubtful that the vast quantities of methane that would be required to trigger substantial warming could reach the atmosphere before being oxidized in the sediments and the overlying water (Archer et al., 2000; Archer, 2007). The search for
mechanisms that may explain the glacial-interglacial variation in atmospheric $pCO_2$ has therefore focused on other scenarios. Common to many scenarios is the idea that the whole ocean inventory of major nutrients may have varied on glacial-interglacial time scales: If the nutrient inventory in the global ocean were to increase, mixing and upwelling could increase the flux of nutrients to the surface ocean. This would stimulate primary productivity, increase the flux of organic matter to the seafloor, and lower the $pCO_2$ in the surface ocean. This scenario is supported by reports that global rate of organic matter
burial is maximal during glaciation (Cartapanis et al., 2016; Boyle, 1986; Wallmann, 2010). If the $pCO_2$ in the surface ocean were to fall below the atmospheric $pCO_2$, $CO_2$ would be drawn down from the atmosphere. A temporally variable nutrient inventory could thus explain the glacial-interglacial $pCO_2$ difference. The combination of processes involved in this scenario has become known as the biological pump (e.g., Falkowski, 1997; Hain et al., 2014).

Searching for a temporary sink for phosphorus, the nutrient often considered to be limiting on geological time scales in the
ocean, Broecker (1982a, 1982b) developed a scenario whereby phosphorus is removed from the global ocean by deposition of P-containing sediment on continental shelves during sea-level high stands and returned to the ocean by erosion of shelf sediments during sea-level low stands. This scenario, known as the "shelf nutrient hypothesis", suffers a major weakness: The amount of sediment that has to be deposited and eroded on the relevant time scale appears to be far greater than can be supported by observations (Peacock et al., 2006). Another weakness of the hypothesis is the absence of an explicit

mechanism that would release phosphate into the aqueous phase once the sediments are eroded. Filippelli et al. (2007) proposed that, with a proper lag time, subaerial weathering by acidic rainfall and plant colonization of exposed shelf sediments will release phosphorus to the ocean. As Archer et al. (2000) stated, it seems that all of the simple mechanisms for lowering pCO$_2$ in the surface ocean have been eliminated. Broecker (2018) made the point that although alternate scenarios may exist to explain glacial cycles, one must take into account that the situation is complex, and that no single scenario may be the 'best'.

## 2 Objectives and approach

In a recent modelling study, we drew attention to the fact that methane is a fuel that can drive diagenesis in sediments. Using manganese as an example, we showed that methane-fuelled diagenesis can reduce and redistribute solid-phase oxidized manganese throughout the sediment column (Sundby et al., 2015). Building on that study, we now focus on methane-fuelled reduction of oxidized sedimentary iron-phases and the consequent release of iron oxide-bound phosphorus into the pore water. The objective is to develop a chronological scenario of phosphorus cycling in the global ocean over a full glacial cycle. In this scenario, which capitalizes on the two-reservoir model of Broecker (1982a, 1982b), phosphorus cycles between two principal reservoirs: the ocean and the sediments that accumulate on continental margins.

We focus on the mechanisms that promote the release of sedimentary phosphorus into the pore waters and the transfer of phosphorus to the oceanic reservoir. We then explore the mechanisms that return phosphorus to the sedimentary reservoir throughout a glacial cycle. Finally, we examine the interaction of the glacial-interglacial phosphorus cycle with the carbon system to discover how this interaction might affect variations in atmospheric pCO$_2$. The approach focuses on large scale temporal and spatial processes, but it does not preclude eventual incorporation of shorter timescale and broader spatial-scale processes in a more detailed scenario. By including diagenesis, we can revive elements of Broecker's original hypothesis by providing a set of mechanisms that do not rely on large scale erosion to release phosphate from sediments. By invoking diagenesis (phosphate released to the pore waters upon the reductive dissolution of iron oxides by sulfide produced during anaerobic methane oxidation), we can show that methane-driven diagenesis may contribute to the redistribution of phosphate between the oceanic and the sedimentary reservoirs.

## 3 Diagenesis and fluxes of phosphorus in continental margin sediments

### 3.1 Sources and sinks of phosphorus

The inventory of the oceanic phosphorus reservoir depends on the supply of phosphorus from the continents, remineralization of organic phosphorus in the water column and the upper sediment column, burial of solid phases with which phosphorus is associated, release of soluble phosphorus from the sediments, and removal by hydrothermal activity (e.g., Wallmann, 2010). Burial on the abyssal seafloor is the ultimate sink for phosphorus, but an amount equivalent to about

half of the total phosphorus flux from the continents to the modern ocean does not reach the abyssal seafloor but settles out and is buried on continental margins (Colman and Holland, 2000; Ruttenberg, 2003). The particulate phosphorus that is delivered to the seafloor contains, on average, one third each of organic phosphorus, iron oxide-bound phosphorus, and poorly-reactive apatite minerals (Delaney, 1998; Fillipelli, 1997; Ruttenberg, 2003). Organic phosphorus exists in a variety of forms (primarily phosphate esters) that originate from excretion, decomposition, death, and autolysis of organisms. Microbial degradation of organic matter during the earliest stages of diagenesis converts organic phosphorus to dissolved inorganic phosphate. Part of the remineralized phosphorus escapes into the water column and is added to the oceanic phosphorus reservoir; part of it is co-precipitated with or adsorbed onto iron (hydr)oxide minerals or may be converted to poorly reactive phases such as calcium fluoroapatite (Delaney, 1998).

In its simplest expression, the marine phosphorus cycle consists of a source and a sink of P-bearing material and a set of processes that act upon this material (Fig. 1). Erosion and weathering of continental rocks supply phosphorus to the ocean, and burial on the seafloor removes it. Until recently, the phosphorus sink was thought to be represented by poorly reactive (stable) apatite minerals (Egger et al., 2015). This view had presumably its origin in the analytical procedures that were used to quantify individual P-containing components in marine sediments. This view is now changing, and other phosphorus minerals are thought to be involved. This is exemplified by the authigenesis of vivianite, a hydrous ferrous phosphate that can form in sulfate-poor sediments such as those in brackish estuaries and may act as an important burial sink for phosphorus in brackish environments worldwide (Egger et al., 2015). Vivianite authigenesis requires pore water with elevated ferrous iron ($Fe^{2+}$) and phosphate concentrations (e.g., Liu et al., 2018). Low-sulfate lacustrine settings typically satisfy these criteria, and the presence of vivianite is commonly reported in freshwater sediments (e.g., Rothe et al., 2014).

In more sulfate-rich settings, the presence of a sulfate–methane transition zone (SMT) has been shown to provide favourable conditions for vivianite authigenesis (Egger et al., 2015, 2016; Hsu et al., 2014; März et al., 2008, 2018; Slomp et al., 2013). The production of dissolved sulfide by anaerobic oxidation of methane and the associated conversion of Fe-oxides to Fe-sulfides, result in elevated pore-water phosphate concentrations around the SMT (März et al., 2008). The subsequent downward diffusion of phosphate into sulfide-depleted pore water below the SMT may lead to the precipitation of vivianite if sufficient reduced Fe is available (Egger et al., 2015; März et al., 2018).

According to the depositional setting, pore-water phosphate profiles in marine sediments tend to display a concentration maximum below the dissimilatory iron and sulfate reduction zone (e.g., Krom and Berner, 1981), within or near the sulfate-methane transition zone (e.g., März et al., 2008) as well as at greater depths where it is most likely released by deep subsurface organic carbon degradation (e.g., Niewöhner et al., 1998). The associated concentration gradients drive vertical transport of soluble phosphate both in the downward and the upward directions. To sustain these fluxes requires the presence of phosphate sinks both below as well as above the SMT zone. This conclusion is in agreement with the postulated presence of a phosphate sink and its strength, but it does not provide information about the nature of the sink.

A large body of research shows that phosphate is in fact released from continental margin sediments into the overlying oceanic water. Perhaps the most prominent results of this research are published in the paper by Colman and Holland (2000).

They analysed nearly 200 published pore-water profiles of dissolved phosphate in sediment cores from a variety of marine
environments and calculated phosphate fluxes into the overlying water. They concluded that the return flux of phosphate
from continental margin sediments was more than one order of magnitude larger than the riverine flux of total dissolved
phosphorus to the ocean. Hensen et al. (1998) reported phosphate fluxes that are slightly lower but of the same order of
magnitude from deep (> 1000m) sediments of the Southern Atlantic Ocean. Wallmann (2010) used a mass balance approach
to estimate phosphate fluxes in the ocean and came up with the remarkable conclusion that the pre-human, modern ocean
was losing dissolved phosphate (to the sedimentary reservoir) at a rate of about 5% $Kyr^{-1}$. Studies of phosphorus diagenesis
in continental margin sediments have also shown that phosphate is released to the overlying waters (Sundby et al., 1992).
These observations point to the important role that diagenesis (oxidation of reduced carbon) plays in the marine phosphorus
cycle. Methane being prominently present in continental margin sediments suggests that our scenario, whereby sea-level
variations have important consequences for methane oxidation and, consequently the marine phosphorus cycle, has merit.

**3.2 Phosphorus diagenesis**

The term *phosphorus* usually stands for the sum total of all phosphorus species, organic as well as inorganic, particulate as
well as dissolved. It is typically defined operationally according to the analytical methods by which it is determined (e.g.,
Ruttenberg, 2003). Phosphorus bound to iron oxides is defined operationally as "reactive particulate phosphorus" but the
latter definition typically includes particulate organic phosphorus, phosphate reversibly adsorbed to other mineral surfaces,
as well as phosphate in carbonate fluorapatite minerals and fish bones. In sediment pore waters, soluble reactive phosphate
(also referred to as SRP, orthophosphate or dissolved inorganic phosphorus) is a minor component of the reactive
phosphorus pool and occurs chiefly as the hydrogen phosphate species ($HPO_4^{2-}$). Particulate and soluble forms of phosphorus
are subjected to different modes of transport. Particulate phosphorus is transported to the seafloor by settling through the
oceanic water column and is buried in the sediment. Soluble forms of phosphorus are transported by diffusion along
concentration gradients that often develop in sediment pore waters and across the sediment-water interface. Transport by
bioturbation and bioirrigation can be important in the upper sediment column.

In this study, we refer to soluble inorganic phosphorus as "*phosphat*e or SRP" and to insoluble forms as "*particulate
phosphorus*" or "*total phosphorus*". The term *reactive iron* has been applied to the fraction of iron in marine sediments that
reacts readily with sulfide (Canfield, 1989). By analogy, we define *reactive particulate phosphorus* as the fraction of total P
that is adsorbed on or is co-precipitated with iron oxides and can therefore be released to pore waters when iron oxides are
reductively dissolved.

The challenge presented by the original shelf nutrient scenario is to find mechanisms that can trigger reciprocal changes in
the sedimentary and oceanic phosphorus inventories over a glacial cycle. Diagenesis is a strong candidate for such a
mechanism. Diagenesis is defined as the sum of the physical, chemical, and biological processes that bring about changes in
a sediment subsequent to deposition (Berner, 1980). Phosphorus diagenesis is intimately linked to the diagenesis of iron
oxide minerals, as these are the most important carrier phases for reactive phosphorus in sediments (Ruttenberg, 2003).

Surface sediments contain both detrital and authigenic forms of oxidized iron, of which the most reactive fractions are typically poorly crystalline (e.g., ferrihydrite and nano-particulate goethite) and occur in the authigenic fraction. Iron oxides have been characterized operationally by their reactivity towards hydrogen sulfide (Canfield, 1989; Roberts, 2015), and various sequential extraction schemes have been designed and applied to distinguish the reactivity of iron oxides towards a variety of reductants (Anschutz et al., 1998; Kostka and Luther, 1994; Poulton and Canfield, 2005; Ruttenberg, 2003). For a recent review of the extensive literature on sequential extraction procedures, we refer to Anschutz and Deborde (2016). Authigenic iron oxides form in the upper part of the sediment column above the depth where the stability boundary for the $Fe(II)/Fe(III)$ redox couple is located. Reactive iron buried below this boundary is reduced to soluble $Fe(II)$ at a rate that depends on the reactivity of $Fe(III)$ phases and organic matter, the availability of which typically decreases with depth below the sediment surface, as well as sedimentation rate and sulfide exposure time (Canfield, 1989). $Fe(II)$ is then immobilized as sulfides (in organic-rich sediment) or is transported up by diffusion across the $Fe(II)/Fe(III)$ redox boundary and reoxidized by various oxidants ($O_2$, $NO_3^-$, $MnO_2$). This "freshly precipitated" iron oxide is very reactive and can be recycled multiple times across the $Fe(II)/Fe(III)$ redox boundary before it is finally permanently buried (Canfield et al., 1993). The presence of authigenic iron oxides in the sediment can be quantified according to its reactivity towards a "weak" reductant such as buffered ascorbic acid (Hyacinthe and Van Cappellen, 2004; Kostka and Luther, 1994). Once buried, authigenic iron oxides may undergo an aging process that includes stepwise dehydration of amorphous phases such as ferrihydrite to goethite and diminishing specific surface area (Lijklema, 1977). This renders reactive iron oxides more refractory which, in addition to diagenetic remobilization, is reflected by a diminishing content of ascorbate-extractible iron with depth in the sediment (Anschutz et al., 1998). Thus, the bulk reactivity of the sedimentary iron oxides typically decreases with time (depth of burial). The reactivity of the sedimentary organic matter that survives burial below the oxic surface sediment is also important because the reduction rate of sulfate and production rate of $H_2S$ depend on it. In the section that follows, we will introduce methane, a highly mobile form of organic matter that can be oxidized anaerobically in sediments by micro-organisms that use sulfate as terminal electron acceptor.

### 3.3 Linking phosphorus diagenesis to sea-level changes via anaerobic methane oxidation

Much of our understanding of the early stages of diagenesis rests on the notion that diagenesis is fuelled by the organic carbon that settles to the seafloor and is buried, and that diagenesis comes to an end when this carbon has been fully consumed. However, in continental margin sediments, where immense sub-surface accumulations of methane are present in the form of methane hydrate (Bohrmann and Torres, 2016; Ruppel and Kessler, 2017), methane can support diagenesis above and beyond that fuelled by organic matter settling from the water column (Burdige and Komada, 2011, 2013; Komada et al., 2016). Methane fuels diagenesis via a number of microbially-mediated processes of which the most important is anaerobic microbial oxidation of methane (AOM) using sulfate as terminal electron acceptor (Boetius et al., 2000; Jørgensen and Kasten, 2006; Nauhaus et al., 2002). AOM has been described as a process that intercepts methane that migrates through anoxic pore water, thereby preventing significant quantities of methane from reaching the atmosphere (e.g., Geprägs et al.,

2016). AOM coupled to sulfate reduction does more than just remove methane; it also produces sulfide and bicarbonate (Eq. 1).

$$CH_4 + SO_4^{2-} \rightarrow HCO_3^- + HS^- + H_2O \qquad (1)$$

The sediment layer within which AOM takes place is known as the sulfate-methane transition zone (SMT). It can be located at depths varying from centimeters to tens of meters below the sediment surface (Borowski et al., 1999). Riedinger et al. (2014) have made the case for coupled anaerobic oxidation of methane via iron oxide reduction, and Egger et al. (2017) have proposed that a potential coupling between iron oxide reduction and methane oxidation likely affects iron cycling and related biogeochemical processes such as burial of phosphorus. Of special interest here is the production of sulfide, a strong reductant that can reductively dissolve even fairly refractory iron oxides and thereby release the associated phosphate into the sediment pore water.

At steady state, the SMT would be located at a fixed distance from the seafloor, and the flux of soluble phosphate from the SMT would be controlled by the burial rate of reactive phosphorus. However, the depth of the SMT in continental margin and epicontinental sediments is not necessarily at steady state on glacial time scales (Henkel et al., 2012). Meister et al. (2007, 2008) linked diagenetic dolomite formation in hemipelagic sediments to sea-level changes, showing that the SMT migrates up and down within the sediment column during the course of 100 Kyr cycles. They hypothesized that the SMT persists within an organic carbon-rich interglacial sediment layer sandwiched between layers of organic carbon-poor glacial sediment. Contreras et al. (2013) reported evidence that the SMT in organic carbon-rich sediments from the highly productive Peruvian shelf has migrated vertically in response to cyclic variations of the carbon flux during glacial-interglacial periods. The depth of the SMT may also fluctuate and the upward flux of methane increase in response to over-pressuring of the underlying gas reservoir, earthquakes or sediment mass movements (Henkel et al., 2011; Fischer et al., 2013)

The upward displacement and ultimate location of the SMT in the sediment column is critical since the reduction of iron oxides by sulfide occurs at this location and within the displacement interval and releases phosphate to the pore water. The closer the SMT migrates to the sediment surface, the greater is the instantaneous flux of sulfate into the sediment and the shorter is the path that phosphate travels before it escapes the sediment. The upward displacement of the SMT and the instantaneous flux of phosphate to the ocean can be linked to the rate of sea-level fall if we assume that a time-variable supply of methane from methane hydrate-bearing sediment layers can also bring about a fluctuating SMT.

### 3.4 Background (steady - state) fluxes of methane and phosphate in methane hydrate bearing sediment

The depth distribution of methane hydrate in sediments is constrained by the relatively narrow pressure and temperature range within which methane hydrate is stable and can exist as a solid (e.g., Ruppel and Kessler, 2017). The lower and upper boundaries of the methane hydrate stability field respond to the accumulation and burial of successive layers of fresh sediment by migrating upward towards the seafloor. Methane hydrate located at the lower stability boundary is thereby

transported below this boundary, where it becomes unstable, dissociates, and releases methane into the pore water. Consequently, the downward directed burial flux of sediment is accompanied by an upward directed flux of methane.

The lower stability boundary for methane hydrate can be observed seismically as a discontinuity in sound transmission as a gas-rich layer develops. High-resolution seismic data from the Blake Ridge crest (Borowski et al., 1999) show that methane gas, released by hydrate dissociation at the lower stability boundary, can be injected well into the overlying hydrate stability zone. The data indicate that methane, in the form of free gas, can migrate hundreds of meters through the hydrate stability zone before re-forming as solid phase hydrate (Gorman et al., 2002). The ability of gaseous methane to migrate across a thermodynamic regime where it should be trapped as a hydrate suggests that gas migration through the sediment column plays an important role in the interaction of sub-seafloor methane with the overlying ocean. Under steady-state conditions, the upper boundary of the methane hydrate stability field would keep pace with the rate of sediment accumulation, which would create a zone in the upper sediment column within which methane can be removed from the pore water by solid phase hydrate precipitation. In the presence of sulfate, which is in abundant supply in the overlying ocean, methane can also be removed from the pore water by anaerobic methane oxidation. According to the stoichiometry of AOM (Eq. 1), the upward flux of methane into the SMT should equal the downward flux of sulfate. In the absence of electron donors other than methane, a linear pore-water sulfate gradient is expected and can be used to estimate the instantaneous methane flux (Borowski et al., 1999).

### 3.5 Sea-level changes and methane and phosphate fluxes in the sediment

Because of the sensitivity of methane hydrate deposits to changes in temperature and pressure, one can expect that fluctuations in sea level and temperature, both of which are associated with glacial cycles, will perturb the depth distribution of methane in the sediment. For example, the hydrostatic pressure decrease that was associated with the 120 m sea-level drop during the last glacial maximum has been estimated to lower the hydrostatic pressure enough to raise the lower boundary of the gas hydrate stability field by as much as 20 m (Dillon and Paull, 1983). There can therefore be little doubt that a perturbation such as a sea-level drop can cause methane to be released into the pore water and thereby increase the instantaneous fluxes of methane and phosphate above and beyond the slower background fluxes sustained by steady-state sediment accumulation (section 3.4).

The stoichiometry of anaerobic methane oxidation requires that the flux of sulfate into the sediment must increase in order to accommodate an increased upward flux of methane (Eq. 1). This implies that the sulfate gradient grows steeper and that the SMT migrates upward (Fig. 2). As the SMT moves upward, it is accompanied by a front of dissolved sulfide produced by AOM that can reduce buried phosphate-bearing Fe-oxides if present. Iron-oxide reduction and dissolved ferrous iron production may generate a downward directed flux of Fe(II) into the sediment located below the SMT. It has been suggested (März et al., 2018) that this might be one of the conditions (among others) that would lead to authigenesis of Fe(II) minerals such as vivianite.

Recent syntheses of global sea-level records (Foster and Rohling, 2013; Lambec et al., 2014; Wallmann et al., 2016) show that there were several episodes of sea-level fall during the last glaciation, each episode interrupted by periods of stable sea level. This suggests that there could also have been several pulses of methane release into the pore water, which would cause the sub-bottom depth of the SMT and the reactions associated with it to fluctuate.

The phosphate flux cannot be expected to mirror directly the fluxes of methane and sulfate because the reductive dissolution rate of the sedimentary iron oxide pool is not constant but depends on the reactivity of the individual iron oxides in the sediment (Canfield, 1989). Furthermore, the efflux of phosphate is ultimately limited by the pool of reactive phosphorus present within the displacement interval of the SMT in the sediment column when sea level falls. If phosphate-bearing iron oxides are absent within the displacement interval (either they never accumulated or were reduced by a previous migration of the SMT) then the phosphate concentration gradient would not be altered significantly, as would the flux of phosphate across the SWI. Even if phosphate-bearing iron oxides are present within the displacement interval, the phosphate gradient and flux across the SWI will wane with time as the phosphate-bearing iron oxides are deactivated (by FeS coating or Fe(II) adsorption) or exhausted (reduced by AOM-generated sulfide). Hence, if the position of the SMT has been invariant for a while, despite a persistent methane flux and/or the presence of methane seeps, we would not expect a significant phosphate flux out of the sediment because phosphate-bearing oxides would either not accumulate in these sediments or have long been dissolved by the sulfidic pore waters, as observed by Niewöhner et al. (1998) in sediments of the upwelling area off Namibia. In fact, pore-water profiles would likely mirror those reported by Wunder et al. (2021) in the Church Trough sediments of South Georgia where cold methane seeps are documented and where little phosphate escapes the sediment. In other words, one would not expect to observe strong phosphate fluxes across the SWI where persistent cold methane seeps are currently found.

### 3.6 Transport of phosphate in sediment pore waters: adsorption and desorption

Relative to the sediment surface, burial moves reactive particulate phosphorus downward into the sediment column while diffusion moves soluble phosphate upwards. The two oppositely directed phosphorus fluxes interact by partitioning soluble phosphate between solution and sorption sites on solid surfaces, mostly to iron oxides (Krom and Berner, 1980, 1981; Sundby et al., 1992). Phosphate also adsorbs onto other solids, including Mn-oxides and carbonate minerals (Millero et al., 2001; Yao and Millero, 1996). It can also form authigenic carbonate fluorapatite and vivianite (März et al., 2018). It has been observed that all the iron oxide surfaces found in oxic continental margin sediments are "saturated" with phosphate (all available sorption sites are occupied) and that the detrital iron oxide fraction is already saturated in phosphate by the time it arrives on the seafloor (Anschutz and Chaillou, 2009). Irrespective, detrital and diagenetic iron oxides in organic-rich sediments have very high but sliding buffering adsorption capacities (Sundby et al., 1992). Hence, if the pore-water phosphate flux to the SWI is increased, much of the phosphate will be intercepted by these oxides, but the concentration of maximum buffering capacity of the sediment (or zero equilibrium phosphate concentration ($EPC_0$), a concept first introduced by Froelich (1998)) will increase and the concentration gradient and flux of phosphate across the SWI will also increase.

Sorption should therefore not fully restrict the transport of phosphate diffusing up from the SMT towards and across the sediment-water interface.

Evidence that soluble phosphate can be transported over large depth intervals in sediments without the pore-water profile being visibly affected by sorption onto sediment particles can be found in the dataset of Niewöhner et al. (1998). Four 12-15 m long cores collected in 1300-2000 m water depth from gas hydrate-bearing sediments on the continental margin off Namibia display linear pore-water phosphate profiles. In two of the cores, the soluble phosphate profile is linear over a 15 m depth interval, from the sediment-water interface to the bottom of the cores; the two other cores display a slope change at about 2 m depth, below which the profile is linear. We do not wish to imply that these profiles are representative of glacial sediments, but they do illustrate an important point: It is possible for soluble phosphate to diffuse over long distances in sediment pore water without encountering significant impediment by secondary diagenetic reactions.

### 3.7 Quantification of methane-fuelled phosphorus diagenesis

Methane hydrate is thought to exist in the pore spaces of marine sediment located in water depths ranging from 500 – 4000 m (Paull et al., 1991; Bratton, 1999). This includes sediment located 0–600 m below the seafloor. Bratton (1999) estimated that 40% to 75% of the total surface area of continental slopes, i.e., 22-41 × $10^6$ km$^2$, contain gas hydrates. The fraction that could be affected by methane hydrate destabilization in response to sea-level fall is located between 500 and 2000 m water depth (Kennett et al., 2003), i.e., 23 × $10^6$ km$^2$ (Costello et al., 2015).

Destabilization of methane hydrate in the sediment column increases the local rate of sulfate reduction as well as the flux of sulfate into the sulfate-methane transition zone (SMT). The depth interval of sediment over which the SMT initially migrates may contain phosphate-bearing refractory iron oxides that were buried below the zone where catabolic processes are fuelled by the organic carbon delivered to the sediment surface. The concentration of iron-bound P located several decimetres or several meters below the water-sediment interface is on average 2 µmol g$^{-1}$ (Colman and Holland, 2000; Slomp et al., 1996; Anschutz et al., 1998; Ruttenberg, 2014). Given a mean porosity of 0.6 for silty continental slope sediments buried several meters below the seafloor (e.g., Charbonnier et al., 2019; Schulz and Zabel, 2006), and a bulk dry sediment density of 2.65 (Berner, 1980), the mass of particles in 1 m$^3$ of wet sediment is 0.4 m$^3$ × 2650 kg m$^{-3}$ or about 1000 kg. This mass of sediment would contain 2 moles of Fe-bound phosphorus, which means that a 1-m thick sediment pile can potentially release 2 moles of P per m$^2$ or 2 Mmol km$^{-2}$. When scaled to the surface area of continental slope sediments located between 500 to 2000 m water depth, this represents a standing stock of 46×$10^{12}$ moles of mobilizable P per meter of sediment thickness.

When the SMT rises during a glacial episode, the maximum thickness of the sediment layer that can release Fe-bound P is equivalent to that corresponding to deposits that accumulated over a 100,000-year glacial cycle. If the vertical migration of the SMT affects more than 100,000 years of sedimentary deposits, a portion of these deposits is then affected by the migrations triggered by two successive glacial cycles. Irrespective, sediment can only release its Fe-bound P once. On the slope, the mean sedimentation rate is estimated at between 15.3 cm Kyr$^{-1}$ (Burwiks et al., 2011) and 42.6 cm Kyr$^{-1}$ (Egger et

al., 2018). Therefore, 100,000 years of sediment accumulation corresponds to a sedimentary column that is 15 to 42 m thick, containing between $0.69 \times 10^{15}$ and $1.93 \times 10^{15}$ moles of Fe-bound phosphorus.

The inventory of dissolved phosphate in the current ocean is estimated at between $2.6 \times 10^{15}$ moles and $3 \times 10^{15}$ moles (Colman and Holland, 2000; Sarmiento and Gruber, 2006). The inventory of iron-bound P in 100,000 years of sediment accumulation thus represents 23% to 76% of the ocean dissolved phosphate inventory. If this inventory is remobilized over the glacial period of 90,000 years, it would correspond to a flux of 0.8 to $2.1 \times 10^{10}$ mol yr$^{-1}$. These values are of the same order of magnitude as pre-agricultural, riverine dissolved phosphate fluxes to the ocean ($1.3 \times 10^{10}$ mol yr$^{-1}$, Meybeck, 1982). This is the first time that a quantitative estimate of the magnitude of the sedimentary mobilizable phosphorus reservoir has been made. A comparison of the oceanic P-reservoir to the sedimentary reservoir reveals that the two reservoirs are of comparable magnitude. This raises the role of methane-fuelled phosphorus from a somewhat speculative idea to a potentially major player in the glacial-interglacial phosphorus cycle. The prospective flux of phosphate fed by the ascent of the SMT is significant if the diffusional transport of P is sufficient for this phosphate to reach the sediment surface. The flux equation for transport via molecular diffusion is:

$$J = -\varphi D_s \, (\Delta(SRP)/\Delta z) \tag{2}$$

where J is the phosphate flux across the sediment-water interface; $\varphi$ is the mean porosity of the sediment; z is the depth coordinate; $(\Delta(SRP)/\Delta z)$ is the phosphate concentration gradient. $D_s$ is the bulk sediment molecular diffusion coefficient, assumed to be equal to $D_s = D_o/(1-\ln \varphi^2)$ (Boudreau, 1996), where $D_o$ is the diffusion coefficient in water at in situ temperature. Considering a porosity of 0.65 and $D_o$ of 0.0124 m$^2$ yr$^{-1}$ for HPO$_4^{2-}$ at 5°C (Schulz and Zabel, 2006), a mean flux of $1 \times 10^{10}$ mol yr$^{-1}$ from a sediment surface of $23 \times 10^6$ km$^2$ (=$4.35 \times 10^{-4}$ mol m$^{-2}$ yr$^{-1}$) requires a concentration gradient $(\Delta(SRP)/\Delta z) = J/(-\varphi D_s)$ of 0.100 mol m$^{-3}$ m$^{-1}$ or 100 µmol L$^{-1}$ m$^{-1}$. This gradient is only 2 to 4 times higher than the highest gradient observed on continental slope sediments from long gravity cores by Niewöhner et al. (1998) and Charbonnier et al. (2019), suggesting that molecular diffusion is not an obstacle to the transfer of the inventory of mobilizable phosphate toward the sediment-water interface, as long as the concentration gradient becomes high, as our scenario predicts. A mean diffusive SRP-flux of $4.35 \times 10^{-4}$ mol m$^{-2}$ yr$^{-1}$ is lower than the phosphate return flux fuelled by organic matter regeneration at the sediment-water interface in continental margins (Hensen et al., 1998; Colman and Holland, 2000). However, the increased flux of SRP from below could increase the pore-water concentration gradient immediately below the sediment-water interface (see section 3.6 and Fig. 2 for explanations).

Methane-fuelled P diagenesis thus adds another trigger to the suite of processes that may explain the increased P flux to the ocean during glacial periods, such as the release of P from the continental margin (Filippelli et al., 2007; Filippelli, 2008; Tsandev et al., 2008), the diagenetic P-fluxes driven by the early diagenetic microbial degradation of settling organic matter, and the decreased P burial efficiency (Wallman, 2003; Palastanga et al., 2013). The scenario we have

developed provides another element to the suite of processes that many have contributed to the rapid glacial-interglacial climate transitions documented in paleo records.

## 4 Linking the phosphorus cycle to changes in atmospheric $CO_2$

### 4.1 The concept of a limiting nutrient

A nutrient is limiting if its addition to the system increases the rate of net primary production. Because phosphorus and nitrogen both are essential elements for life, there has been much debate about which of these elements limits photosynthetic primary production in the ocean (Falkowski, 1997; Galbraith et al., 2008; Lenton and Watson, 2000; Smith, 1984; Tyrrell, 1999). Oceanic inventories of nitrogen underwent large changes between glacial and interglacial periods (Ganeshram et al., 1995, 2002). They have been attributed to greatly diminished water column denitrification and consequent increase in the nitrate inventory during glacial periods. In the modern ocean, it appears that nitrogen is the limiting nutrient. Therefore, increasing the flux of phosphate into the ocean would not necessarily increase the rate of primary production. Nevertheless, irrespective of which nutrient element is limiting, primary production assimilates phosphate into biomass, which settles to the seafloor and is buried under successive layers of new sediment. In this way, phosphorus is transferred from the oceanic reservoir to the sedimentary reservoir. Likewise, the net result of methane-fuelled diagenesis is to return phosphate, a nutrient element, from the sedimentary to the oceanic phosphorus reservoir.

### 4.2 Expansion and contraction of phosphorus reservoirs during a glacial cycle

The central idea of the simple two-reservoir representation of the phosphorus cycle proposed by Broecker (1982a, 1982b) is that the phosphorus inventory in a reservoir can contract and expand on glacial time scales, eventually impacting the $CO_2$ level in the atmosphere. This simple representation of otherwise complex phenomena has stimulated research on the mechanism that control nutrient fluxes in the ocean.

Boyle (1986) showed that the oceanic phosphorus inventory can vary on a glacial time scale. He concluded, based on the cadmium content of foraminifera and carbon isotope measurements, that the phosphate content of the ocean during the last glacial maximum was 17% larger than it is at present. Likewise, Wallmann (2010) found that the pre-human modern ocean was losing dissolved phosphate (to the sedimentary P-reservoir) at a rate of about 5% $Kyr^{-1}$. The 'lost' phosphate is assimilated into biomass and/or adsorbed onto mineral particles, the settling of which returns reactive phosphorus to the sediment. Colman and Holland (2000), who examined phosphate cycling in modern continental margin sediments, concluded that the current efflux of phosphate from these sediments is about one half of the total settling flux of particulate reactive phosphorus. The portion that settles through the ocean water column to the seafloor is eventually converted to stable minerals and lost from the oceanic phosphorus cycle.

### 4.3 Chronology of events during a glacial phosphorus cycle

With the initiation of a glacial cycle, the global temperature decreases, ice builds up on the continents, the sea-level falls, the pressure on the seafloor decreases, and the lower boundary of the methane hydrate stability field in the sediment column shifts upward. Methane hydrate located below the new lower stability boundary becomes unstable and decomposes. Methane gas is released to the pore water, the concentration gradient becomes steeper, and the upward methane flux increases. Upward migrating methane encounters downward diffusing sulfate in the sulfate-methane transition zone (SMT). Here, methane is oxidized to $CO_2$ and sulfate is reduced to sulfide (Eq. 1). The reaction between sulfide and ferric minerals, present within the displacement interval of the SMT, reduces Fe(III) to Fe(II), and phosphate associated with these solids is released into the pore waters, increasing the soluble reactive phosphorus concentration gradient and its flux into the oceanic reservoir (Fig. 2) until the ferric minerals are deactivated or exhausted. Irrespective of which nutrient is limiting, primary production in the photic zone assimilates nutrients into biomass, lowering the inventory and concentration of soluble reactive phosphorus and other nutrients in the oceanic reservoir. The fraction of reactive phosphorus that becomes buried on the continental margin is not necessarily lost to the marine phosphorus cycle. It may conceivably become remobilized by diagenesis during an eventual new glacial cycle.

During times when the sea level remains stable, and assuming that sediment accumulation and burial still take place, methane-fuelled diagenesis can nevertheless occur, as the upper and lower boundaries of the methane hydrate stability field track the burial of sediment. Hence, gas hydrates decompose as the lower stability boundary moves up and release methane, and subsequently upon AOM at or slightly above the upper stability boundary, release phosphate to the pore water. Phosphate is also released during the microbial remineralization of organic matter reaching the seafloor and, hence, a flux of phosphate—what we call a 'background' flux— can therefore be delivered to the ocean even when the sea-level remains stable. When sea-level change is then superimposed, the net result is to temporarily amplify the phosphate flux to the ocean beyond the background flux.

Sea level-driven diagenetic transfer of phosphate from sediment to ocean continues throughout the glaciation period, perhaps in spurts because of intervals of stable sea level. When, after the deglaciation, sea level and pressure on the seafloor have reached a new steady state, diagenetically driven fluxes from the sediment stabilize on pre-glaciation background values. Towards the end of the glaciation period, the stratification of the water column destabilizes (e.g., Basak et al., 2008), which facilitates vertical mixing and brings $CO_2$-rich deep water to the photic zone. The biological pump having been weakened by the ongoing biological removal of soluble reactive phosphorus from the oceanic reservoir and the physical pump having gained strength, transport of nutrient-poor/$CO_2$-rich water to the surface ocean will favour the escape (outgassing) of $CO_2$ into the atmosphere. The rise in atmospheric $CO_2$ levels during the glacial period may also have been helped along by the oxidation of iron sulfides in exposed carbonate-rich shelf sediments (Kölling et al., 2019).

# 5 Summary and conclusions

We have modified the two-reservoir scenario of the marine phosphorus cycle proposed by Broecker (1982a, 1982b) to include a diagenetic mechanism that allows for phosphorus to be exchanged between the sedimentary and the oceanic reservoirs on glacial time scales. Within this scenario, a coupled series of processes act upon the glacial – interglacial marine phosphorus cycle (Table 1).

During a glaciation period, water is transferred from the ocean to continental ice sheets. The falling sea level lowers the hydrostatic pressure on the seafloor. The pressure change perturbs the stability field of methane hydrates, and methane gas is released into the pore water. The release of methane gas from gas hydrates amplifies the upward flux of methane through the sediment column. In the sulfate-methane-transition zone (SMT), where methane is removed via anaerobic methane oxidation, the increased methane supply increases the demand for sulfate. The sulfate gradient steepens, and the SMT zone moves closer to the sediment-water interface. Reactions within the SMT produce hydrogen sulfide, which reductively dissolves iron oxides within the displacement interval and near the SMT. Iron-bound phosphate can then be released into the pore water. The upward directed phosphate gradient steepens, which increases the flux of phosphate towards and across the sediment-water interface and increases the phosphate inventory of the ocean.

Phosphorus is transferred from the oceanic to the sedimentary reservoir by sedimentation and burial of phosphorous-containing biogenic particulate matter resulting from primary production and abiotic particulate matter on which phosphate is adsorbed. Unlike the release of phosphate from the sediment, the return flux of phosphorus to the sediment and the associated depletion of the oceanic reactive phosphorus inventory do not occur as a single event but take place throughout the glacial cycle. Burial of organic matter causes a corresponding loss of phosphorus from the oceanic inventory. If phosphorus were the nutrient limiting primary production at the time leading up to the release, primary production would be stimulated by the added phosphate to the surface ocean, $CO_2$ would be drawn down from the atmosphere, and the phosphorus inventory would begin to decrease. If, on the other hand, nitrogen nutrients were to become limiting, which is likely when new phosphate is added to the surface ocean reservoir, the rates of primary production, phosphorus burial, and draw-down of atmospheric $CO_2$ would slow down until the fixed nitrogen to phosphorus ratio became similar to the Redfield ratio. It is therefore not unreasonable to expect that with a gradual depletion of phosphorus in the ocean, a stable N:P ratio would develop, co-limiting primary production in the ocean (Lenton and Watson, 2000; Lenton and Klausmeier, 2007). As the glaciation progresses, the ocean will gradually become nutrient depleted irrespective of which is the limiting nutrient. This would weaken the biological pump relative to the physical pump (upwelling of deep $CO_2$-rich water) and set the stage for a tipping point beyond which atmospheric $CO_2$ is controlled by upwelling of $CO_2$-rich deep water. The next glaciation period, which is accompanied by sea-level fall, methane-hydrate decomposition, anaerobic methane oxidation, and phosphate release allows the biological pump to once again take control over atmospheric $CO_2$.

The proposed scenario capitalizes on elements of Broecker's model and provides another potential trigger to the suite of processes that may have contributed to the rapid glacial-interglacial climate transitions documented in paleo-records.

The central idea of the simple two-reservoir representation of the phosphorus cycle proposed by Broecker (1982a, 1982b) is that the phosphorus inventory in a reservoir can contract and expand on glacial time scales, eventually impacting the $CO_2$ level in the atmosphere. This simple representation of otherwise complex phenomena has stimulated research on the mechanism that control nutrient fluxes in the ocean.

## Data availability

No new data are presented in this manuscript, which is inspired by and developed with data in the public domain. These are cited in the references included in the manuscript.

## Author contribution

BS, PA, PL, and AM designed the study. BS wrote the manuscript with input from all authors.

## Acknowledgements

B.S. and A.M. gratefully acknowledge the financial support from the Natural Sciences and Engineering Research Council of Canada through their Discovery grants. We wish to thank the three journal reviewers, Drs. D. Archer, G. Filippelli and S. Kasten for their incisive comments on a previous version of this manuscript.

## Financial support

This work was supported by the Natural Sciences and Engineering Research Council of Canada (grant number RGPIN/04421–2018)

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

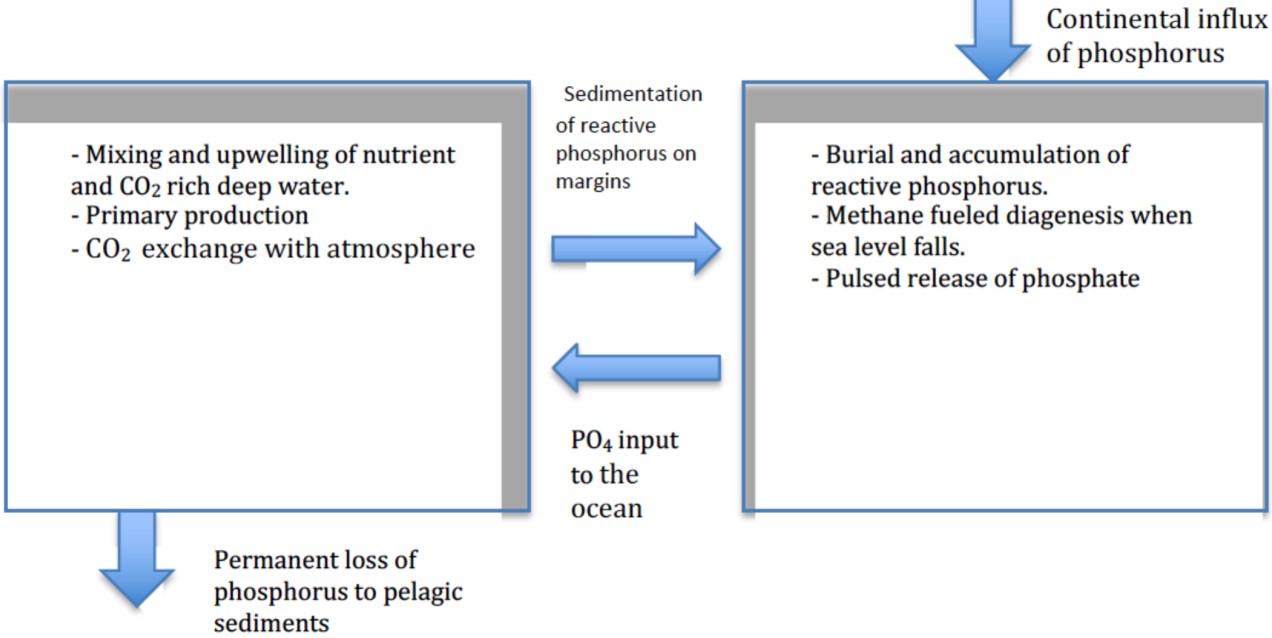

**Figure 1: A simple two-reservoir representation of the oceanic phosphorus cycle.**


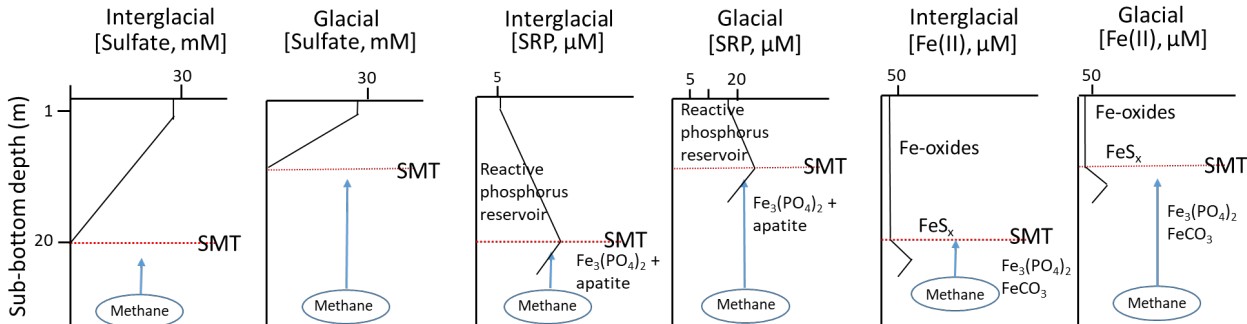

**Figure 2. Pore-water profiles for relevant species (CH$_4$/SO$_4$/Fe(II)/H$_2$S).** Note that the SMT migrates vertically when forced by variable CH$_4$ fluxes. Given the added SRP flux generated by the vertically migrating SMT, the concentration of maximum buffering capacity of the sediment (or zero equilibrium phosphate concentration (EPC$_0$), see Sundby et al. (1992) for details, a concept first introduced by Froelich (1988)), will be rapidly exceeded and the pore-water SRP concentration gradient across the SWI will strengthen. If the flux of SRP from below is large enough, the buffering (or adsorption) capacity of the authigenic iron oxides may be overwhelmed and the linear gradient could extend all the way to the SWI.

**Table 1: The glacial-interglacial phosphorus cycle: Events and consequences.**

| EVENTS | IMPACTS | NOTES |
|---|---|---|
| 1. A glacial cycle begins | Temperature decreases globally, ice builds up on the continents, sea level begins to fall. | |
| 2. The pressure on the seafloor decreases. | The upper and lower boundaries of the methane hydrate stability field in the sediment column shift upward. | |
| 3. Methane hydrate located below the lower stability boundary becomes unstable and decomposes. | Methane is released to the pore water, the methane concentration gradient steepens, and the upward m ethane flux increases. | *Had the sea level remained stable, methane-fuelled diagenesis would nevertheless have taken place and a flux of phosphate released as a result of slowly migrating SMT with sediment accumulation as well as through early diagenesis (remineralization of organic matter reaching the seafloor)—which we call a 'background' flux— would have been delivered to the ocean. The result of sea-level fall is therefore to generate a sudden displacement of the SMT and increase the phosphate flux above and beyond the background.* |
| 4. Upward-diffusing methane encounters downward-diffusing sulfate, which increases the rate of anaerobic methane oxidation (AOM) in the sulfate-methane transition zone (SMT). | The SMT moves up towards the sediment-water interface. | *Methane can also be oxidized using other electron acceptors such as manganese oxides, which would also release adsorbed phosphate to the pore water.* |
| 5. Pore-water hydrogen sulfide accumulates at the displaced SMT | Hydrogen sulfide reduces Fe (III) minerals, if present, to Fe (II), thereby releasing phosphate associated with Fe oxides into the pore water. A pore-water phosphate maximum appears within the displacement interval and close to the SMT, which supports upward and downward fluxes of SRP. Ferrous sulfides precipitate. | *The upper part of the sediment column, located above the methane oxidation zone, contains reactive phosphorus that was delivered and buried during the previous glacial cycle. This accumulated mass of phosphorus is available for conversion to SRP and may be thought of as a limiting factor controlling the expansion of the oceanic phosphorus inventory during a glaciation cycle.* |
| 6. The zone in the sediment column within which phosphate associated with iron oxides can be released expands in the direction of the seafloor. | The SRP concentration gradient steepens and the SRP flux towards the sediment-water interface and into the ocean increases. | *The kinetics of iron oxide reduction by $H_2S$ depends on the reactivity of the oxides, which itself depends on the mineralogy, time since deposition, deactivation of their surfaces, etc. Sequential analysis of sediments suggests that the most reactive forms of* |

| | | *iron oxide occur in the upper part of the sediment column. With time these oxides typically become exhausted or converted to less reactive forms with depth (time) in the sediment.* |
|---|---|---|
| 7. Nutrients are assimilated by primary producers and incorporated in biomass. | Irrespective of which nutrient is limiting, primary production incorporates nutrient elements in organic matter, which settles to the seafloor and is progressively buried. | *The part of the export production that reaches the seafloor in the deep abyssal ocean is oxidized aerobically, and the phosphorus it contains is converted to stable, poorly soluble minerals such as fluorapatite and is removed from the oceanic phosphorus cycle. Reactive phosphorus in the material that settles on the continental margin is available for diagenesis and may be remobilized and returned to the ocean.* |
| 8. SRP is released into the oceanic surface layer, temporarily decoupling the oceanic P and C cycles. | Reduction of Fe oxides in continental margin sediments delivers SRP to the upper water column. Oxidation of organic matter settling through the abyssal water column enriches deep water in $CO_2$. Oxidation in the underlying sediment converts phosphorus to stable phosphorus minerals. | This is supported by observations: global rate of organic matter burial is maximal during the glaciation (Cartapanis et al., 2016; Boyle, 1986; Wallmann, 2010). |
| 9. Stratification of the water column breaks down during the termination of the deglaciation, destabilizes the water column, and facilitates vertical mixing. $CO_2$-rich bottom water is transported to the surface ocean. Degassing releases $CO_2$ to the atmosphere. | $CO_2$-rich, nutrient-poor water is brought to the surface. This weakens the biological pump, which would otherwise draw down $CO_2$ from the atmosphere. The partial pressure difference would be controlled by the supply of $CO_2$ from the deep ocean. The $CO_2$ flux would be directed from the ocean to the atmosphere and create a peak in atmospheric $CO_2$. | |
| 10. Upon termination of the glaciation the sea level rises and increases the pressure on the seafloor. | The fluxes of methane and diagenetically released SRP become weaker and return to pre-glaciation background levels. | |