# Peer review of "Ideas and perspectives: Sea-Level Change, Anaerobic Methane Oxidation, and the Glacial-Interglacial Phosphorus Cycle"

_Biogeosciences, 2021_

## Referee Comment (RC3)

**Review of Sundby et al. , Biogeosciences**

The „Ideas and Perspectives" manuscript by Sundby et al. presents a new potential scenario that adds to the suite of processes that may have contributed to the rapid climate changes observed along glacial/interglacial transitions - namely the documented rapid increase in atmospheric CO2 concentrations during deglacials or more precisely prior to glacial terminations. The authors suggest a mechanism that links sealevel, rates of anaerobic oxidation of methane and benthic fluxes of phosphate in this way contributing a new mechanism to the so-called „shelf nutrient hypothesis" initially developed by Broecker (1982).

I find it very interesting and scientifically fruitful to put forward this scenario and in particular in this way stimulate discussion over a broad spectrum of disciplines. Being a „sediment diagenesis" person myself, I hope you understand that I will mostly focus on related issues.

The authors propose the following suite of conditions and processes: 1) during glacial sea level lowstands hydrostatic pressure is lowered, 2) this enhances the upward flux of methane from underlying gas hydrates and shifts the sulfate/methane transition (SMT) to a shallower depth in the sediment, 3) as a consequence rates of anaerobic oxidation of methane (AOM), hydrogen sulfide formation, reductive dissolution of reactive Fe oxide minerals by sulfide and associated release of adsorbed phosphate into the pore water all increase, 4) this induces higher upward (and downward) fluxes of phosphate, and 5) also increases the flux of phosphate across the sediment/water interface into the bottom water.

While I fully agree with the conditions and processes in points 1 to 4, I am not convinced with the statement given as point 5 above – namely elevated benthic fluxes of phosphate into the water column during a shallower position of the SMT during glacials. Unfortunately the authors have neither presented nor discussed data that demonstrate that phosphate fluxes into the bottow water are indeed higher when the SMT is positioned at shallow sediment depth. They exclusively discuss gravity core data, however, completely neglect the part of a sediment that is key in determining the flux of phosphate across the sediment/water interface – namely the sediment surface. Benthic fluxes can only be assessed if the pore-water gradients of phosphate in the uppermost few decimeters of the sediments are determined. In order to quantify the diffusive flux of phosphate from the sediment back to the bottom/deep water – and thus into the oceanic reservoir - pore-water data of multiple cores or push cores (or benthic chambers) are required, which allow a proper disturbance-free sampling of the sediment surface. Hence, also the schematic representation given in Fig. 2 is not correct, at least as given. With the constant concentrations of phosphate displayed in the uppermost part of the graphs, there is/can be no phosphate flux across the sediment-water interface (SWI) – neither during interglacials nor glacials.

There is also some contradiction and imprecise discussion with respect to the assumed phospate concentrations in the oceanic reservoir/deep waters and the major particulate carrier phases that transport phosphorus into the sediment (organic matter, Fe-bound phosphate) throughout the manscript, which needs to be sharpened and better structured. Moreover, parts of the manuscript contain several incorrect statements and assumptions. concerning the stability of methane gas hydrates and the characterisitics of methane transport both is dissolved and gaseous form in marine sedimentary environments. I was also wondering whether considering issues related to the (potential) release of methane during glacial/interglacial sea level changes really falls into the scope of this manuscript.

The scenario presented also does not consider the processes that occur in exposed shelf sediments during glacial sealevel low-stands. As presented and suggested by Kölling et al. (2019) aeration/oxidation of shelf sediments during sea-level low stands will oxidize reduced iron sulfide minerals resulting in the formation of abundant reactive $Fe(III)$ that will certainly serve as an efficient trap for phosphate either by co-precipitation or adsorption of phospate on $Fe(III)$ mineral surfaces. I would therefore assume, that these (bio)geochemical processes acting in exposed shelf surface sediments are a significant sink for phosphate during glacial periods. How important is this potential sink compared to the (potential) release of phosphate from continental margin sediments into the oceanic reservoir?

So, to conclude, the major issue I have is that the authors have not convingly shown and discussed that benthic $PO_4$ fluxes from continental margin sediments were/are indeed higher during glacial times or in general during times of a shallower position of the SMT – mainly because they have not at all considered and discussed pore-water data for the sediment surface proper. I think that such a scenario could be easily and convincingly tested by comparing sites – i.e. data of MUC or push cores that allow to sample the sediment surface proper - where the SMT is located at different sediments depths. Perhaps also pore-water phosphate data for active seep sites are available/published. In this way the authors could test their hypothesis that during periods of a shallower position of the SMT (and/or periods of active methane seepage) phosphate fluxes across the SWI are/were indeed higher than with a deeper SMT.

I also think that the argumentation can be significantly strengthened by better structuring the manuscript with respect to 1) precisely stating the postulated changes in phosphate concentrations in the oceanic reservoir during glacial and interglacial times, 2) precisely and consistently discussing the main carrier phases of P into marine sediments – namely organic matter and Fe-bound P, and 3) also carefully checking the parts of the manuscipt where you discuss (changes in) hydrate stability and the processes transporting methane through sediments and across the SWI.

**Specific comments and corrections**

Line 13: „additional" to what precisely ?

Ls. 15/15: Here you only mention biomass as a carrier phase to return/carry P back to the sediment. What about Fe-bound P, which – as you state on page 3 – is the key issue of your study/scenario?

Ls. 20/21: This somehow contradicts your statement in lines 13-15 that phosphate fluxes were/are higher during glacial. Why does the deep water then has lower P concentrations during/at the beginning of a deglaciation? I find this very confusing.

Page 2: In this context, I would like to draw your attention to the recent paper by Kölling et al. (2019) who have presented a scenario explaining the rapid increase in atmospheric CO2 concentrations prior to glacial terminations – also linking sealevel changes to atmosphereic CO2 concentrations, more precisely considering high rates of pyrite oxidation on continental shelves exposed during glacial sealevel low-stands.

Kölling, M, Bouimetarhan, I, Bowles, MW, Felis, T, Goldhammer, T, Hinrichs, K-U, Schulz, M, Zabel, M (2019): Consistent CO2 release by pyrite oxidation on continental shelves prior to glacial terminations.- nature geoscience **12**, 929-934. doi 10.1038/s41561-019-0465-9.

L. 46: methane hydrates

Ls. 46 ff: I do not agree with the scenario that sealevel fall-induced lowering of the hydrostatic pressure necessarily leads to a transport of significant amounts of methane into the water column and subsequently into the atmosphere. Of course a lower sealevel will induce a thinning of the gas hydrate stability zone (GHSZ). If methane is transported by diffusion, the methane transported upward from gas hydrates will be more or less completely oxidized/consumed before it can reach the water column. If methane is transported in the form of free gas, these bubbles can reach the lower water column. However, numerous studies have shown that they do not significantly contribute to transport of methane into the atmosphere because methane is rather oxidized areobically in the water column and/or dispersed by horizontal advection and dilution. Moreover, rising bubbles constantly change their internal gas composition due to counterdirected diffusion of gases (methane out, N and CO2 in) across the bubble/water interface (cf. studies by Mc Ginnis or Leifer). As a consequence even in shelf settings with shallow water depth, methane does not reach the atmosphere in considerable amounts (e.g. Mau et al., 2015; Biogeosciences; Geprägs et al., 2016; G3).

Ls. 82 ff.: I also do not fully understand why you discuss the (very controversial) role of methane in the atmospheric carbon cycle over glacial/interglacial changes – please see comments above. Is this really in the scope of your manuscript?!

L. 116: … downward (and upward) diffusion …; into sulfate- and sulfide-depleted pore water below the SMT ….

L. 119: drive instead of imply

Ls. 124 ff.: Please, also check the earlier papers by Hensen et al. in this context:

C Hensen, H Landenberger, M Zabel, HD Schulz (1998) Quantification of diffusive benthic fluxes of nitrate, phosphate, and silicate in the southern Atlantic Ocean. Global Biogeochemical Cycles 12 (1), 193-210.

C Hensen, M Zabel, HN Schulz (2006) Benthic cycling of oxygen, nitrogen and phosphorus. Marine Geochemistry, 207-240.

Ls. 130 ff.: Can you specify? In which kind of environment? Oxygen minimum zones? In this context please also consider the release of Fe-bound P in sediments underlying continental margin oxygen minimum zones – as also typically observed in anoxic lakes (sometimes referred to as internal fertilization)

Ls. 144 ff.: As already stated above, the sediment surface (upper sediment column) is not only important with respect to bioirrigation and bioturbation, but the key locus that determines the diffusive flux of phosphate across the sediment surface into the bottom water.

Ls. 166 ff.: No, Fe2+ generally does not diffuse across the Fe(II)/ Fe(III) redox boundary but is mostly oxidized at this redox boundary by nitrate and generally does not make it further up to the lower boundary of the oxic zone (cf., Froelich et al., 1979; Berner 1981; Kasten et al., 2003).

Ls. 174 ff.: No, the reactivity of sedimentary Fe oxides does not necessarily decrease with time or depth of burial. As numerous studies have shown for shelf and continental margin sediments (Riedinger et al., 2005, GCA; Riedinger et al., 2017, Frontiers in Earth Sciences; März et al., 2008; Oni et al., 2015, Frontiers in Microbiology; Köster et al., 2021, G3) high amounts of reactive Fe (III) minerals can be buried to substantial sediment depth if sedimentation rates are high (as typical for continental margin settings) - thus limiting „sulfide exposure time".

Ls.: 200 ff.: The depth of the SMT may not only fluctuate over glacial/interglacial timescales (e.g. Henkel et al., 2012, GCA) but also be affected by increases/changes in the upward flux of methane induced by overpressuring of the underlying gas reservoir and/or triggered by earth quakes or sediment mass movements (e.g., Fischer et al., 2013, Nat. Geoscience; Henkel et al., 2011, G3; Henkel et al., 2012, Springer book on Submarine Mass Movements).

Ls. 206 ff.: I do not agree with this statement. A shallow SMT does not necessarily mean that the flux of phospate across the SWI is increased. Please, give examples/data of MUC or push cores that demonstrate this.

What do you mean with „instantaneous" flux? This is not clear to me at all.

Ls. 119 ff.: This paragraph on gas hydrate stability and transport of gas containes several flaws and imprecise statements. L. 119: methane is not „produced" by hydrate dissociation but released from the hydrate phase.

The upper boundary of the gas hydrate stability zone (GHSZ) is determined by temperature and water depth/pressure and not by sediment accumulation.

What do you mean with „instantaneous" methane flux? Bubble ebullition, migration of free gas? Please, specify.

Ls. 238-240: Again, what do you mean with „instantaneous" methane flux? I do not at all agree with the statement in this sentence. At least you should give examples and also precisely state which part/interval of the sediment you refer to. The fluxes of both constituents can indeed by higher in the deeper sediments around the SMT but not necessarily at/across the SWI.

Ls. 244-245: No, this is not exactly what we see. There is both a downward flux and an upward flux of Fe2+ towards the SMT (cf., Riedinger et al., 2005, GCA, 2017, Frontiers; März et al., 2008).

L. 249: No, as already stated above, I do not agree that during a shallow location of the SMT the flux of phosphate across the SWI is increased (at least I have seen no data).

Ls. 254 ff.: The benthic phosphate flux is also significantly dependent on the redox/oxyen conditions oft he overlying bottom water. Cf. comment for l. 130.

Ls. 265 ff.: I do not fully agree with the discussion in this paragraph. It may be that Fe(III) phases arriving at the seafloor are already close to saturation with respect to potential sorption sites for phosphate. This does, however, not hold true for the freshly formed/authigenic Fe oxides at the Fe redox boundary. You have also discussed this in previous paragraphs of the manuscript. Numerous data show that most of the upward diffusing phosphate is trapped in the vicinity of the Fe redox boundary, i.e. the pore-water gradient of phosphate changes and only minor amounts of phosphate make it to the overlying bottom water.

Ls. 270 ff.: You are absolutely right that phosphate may diffusively migrate in pore water over relatively large distances of several meters and more – i.e. Niewöhner et al. (1998). This is particularly true for sediments underlying high productivity areas like off Namibia, in which only low amounts of reactive Fe(III) are preserved at depth due to the high rates of sulfate reduction/sulfide production. However, this is certainly not the typical situation in the vast area of continental margin/slope depositional settings. Moreover, please consider that this Niewöhner et al. (1998) paper (and others you have discussed in your manuscript) only shows data for gravity cores. During sediment sampling with gravity cores the uppermost decimeters of the sediments are always lost, so these data do not allow to assess the flux of phosphate across the SWI.

Chapters 3.7 and 4, pages 9 ff.: I do not agree to several of the assumptions presented and discussed here. First, I find it confusing that in your calculation of the sedimentary inventory of P you do not consider organic matter (OM) – although you highlight the OM burial pathway as a key carrier phase of P to the sediment in the abstract and other parts of the manuscript. How much is it compared to the Fe-bound P and how much P is released to the pore water as a consequence of mineralization of OM (compared to reductlive dissolution of Fe(III) minerlas by sulfide? Second, I do not agree that the calculated inventory of Fe-bound phosphate has a chance of ultimately ending up in the water column.

You also have not discussed whether you think that methane transport occurs via diffusion and/or advection – i.e by methane seepage/bubble ebullition. If methane transport mostly occurs via diffusion then both methane and phosphate – although initially released into pore water - will be mostly trapped in the sediments overlying the gas hydrates (methane at the SMT) and phosphate at the Fe redox boundary close to the sediment surface.

From recent studies in continental margin oxygen minimum zones we see that phosphate is only transported from the sediments into the overlying bottom water at high rates under conditions of oxygen-depleted/anoxic bottow waters or at times of active methane seepage – i.e. ebullition of gaseous methane.  – phosphate may be transported at elevated rates into the overlying water column. – in a process similar to mixing of pore water into the bottom water produced by bioturbating/bioirrigating benthic organisms.

If methane transport occurs in the gaseous form – i.e. as bubble ebullition, this occurs along preferential migration pathways, which are spatially (and temporarily) restricted and thus AOM occurring close to theses sites/pathways of gas migration also does not have the capacity to drive reductive Fe(III) reduction over a broad front. Therefore certainly not being able to reductively mobilize the calculated Fe-bound P inventory.

Page 11, upper paragraph: Here, you only speak of OM as a carrier phase to transfer P to the sedimentary reservoir. See also previous comment above.

Chapter 4.3: Here are numerous flaws and imprecise statements with respect to the GHSZ and the transport of methane in marine sediments. A few examples:

Ls. 349/350: No, during sea level drop the upper boundary of the GHSZ moves down (not up). Thus the GHSZ in the sediment gets thinner. There also seems to be some confusion with respect to the upper boundary of the GHSZ and the upper boundary of gas hydrate-bearing sediments. The upper boundary of the GHSZ is (with typical water column temperature and in water depths deeper than about 300 m) found in the water column. However, hydrate formation does not occur in the shallow sediments due to a lack of methane, which is lost to the SMT overlying the gas hydrate-bearing sediments. Methane hydrates also constantly dissolve and release gas from the upper hydrate layers due to the concentration gradient produced by AOM occuring in the overlying SMT. This occurs even if hydrates are well within the hydrate stability zone due to undersaturation of the surrounding pore water with respect to methane (cf. Lapham et al., 2010, EPSL; Kasten et al., 2012, Geo-Marine Lett.).

L. 355: No, as stated above I am not convinced that this necessarily increases the P flux into the oceanic reservoir.

Figure 1: What precisely do you mean with „pulsed" release of phosphate? This is not clear at all and has also not been discussed in the text. Please indicate where and how the two most important particulate carrier phases of P – i.e. OM and Fe-bound P – are transported into the reservoirs.

Figure 2 needs a complete overhaul. The caption of the figure does not correspond to what is shown in the figure (e.g., the profiles of methane and Fe2+ are not shown) and for part of the profiles it is not clear what is shown (what are Fe oxides and what are Fe sulfide minerals?).

Also the schematic representation in this figure does not correspond or represent what the authors discuss. The phosphate profiles shown (seems that they have been adopted from gravity core data of Niewöhner et al. (1998)) have uniform concentrations in the uppermost part of the sedimentary column. This means that there is definitely no diffusive flux of P across the sediment/water interface – neither during interglacials nor glacials.

Table 1: Point 6, column on the right: no, as already outlined above it is definitely not true for continental slope/margin sediments that the most reactive forms of Fe occur in the upper part of the sediment. Please revise and specify.

Point 10, left column: it has to be „increases" instead of „lowers"

I hope that my comments and considerations help focussing the manuscript. It was fun reading and considering your hypothesis/scenario (this is also why my comments are so lengthy ;-) ).

All the best,

Sabine Kasten

---

## Author Response (AR1)

**Compiled Associate Editor and reviewers' comment and responses**
Reviewers' comments are repeated below and are followed by our responses **in bold**

**Associate Editor:** Tina Treude

thank you for submitting your responses to the reviewers. I enjoyed the discussion and I am convinced that you are proposing a very interesting glacial-interglacial mechanisms.

**We thank Dr. Treude for the encouraging words.**

Two of your reviewers had only minor comments, while the review by Dr. Kasten goes into the depth of biogeochemical processes and fluxes of solutes in the sediment.

I agree with Dr. Kasten that some aspects would benefit from clarification, respectively, need to be taken into account. For example, relying on gravity core data is indeed tricky as fluxes into the water column are ultimately determined by the gradient in the sediment layer that connects with the water column - the layer that is lost during gravity corer sampling. I therefore strongly recommend to also reference data from multicorer, pushcorer or benthic flux chamber measurements (where available).

**We are pleased to read that you are convinced that we are proposing very interesting glacial-interglacial mechanisms. We agree with your recommendation to not rely exclusively on data from gravity cores, but use reference data from multi-corer, push-cores or benthic flux chamber studies (where available). We therefore refer to several studies of solute exchange across the sediment-water interface, including a collection of 27 undisturbed box cores from the deep Laurentian Trough in the Gulf of St. Lawrence; as well as in situ benthic chamber studies using divers to specifically avoid disturbing the sediment-water interface. The data obtained in these studies show without a shadow of doubt that phosphate is in fact released from the sediment on seasonal and longer time scales. This information is now included in our revised manuscript. On the other hand, we were not able to find studies of modern cold seep sediments that span glacial-interglacial time scales that would support our scenario.**
**We thank you for bringing this weakness in our scenario to our attention. The published paper will be stronger for it.**

I think it is also worth while looking into phosphate fluxes available from modern methane seep sediments. I understand your point that highest fluxes of phosphate are expected during non-steady state conditions, i.e., when the SMTZ moves closer to the SMI. However, we know that cold seeps are in constant change and hence it is possible that some measurements at seeps reflect such non-steady state conditions.

**We found several references that describe the distribution of phosphorus in mineral concretions associated with cold seeps. However, we unfortunately did not find any publications that describe dissolved phosphorus fluxes in these environments.**

Overall, I believe looking closer into existing datasets could potentially strengthen your hypothesis.

I am looking forward to your revised manuscript, which will be returned to a second round of review.

With kind regards
Tina Treude

**RC1**: Gabriel Filippelli

In this thought-piece, Sundby et al. consider the impact that diagenesis of methane might play in amplifying glacial-interglacial changes via a phosphorus mechanism. Broecker pioneered the concept of climate transition amplification to explain the rapid ~10 ky transition from glacial to interglacial that is reflected in the geologic record. The climate forcing would indicate a sine curve transition, but the record shows that there are significant internal earth mechanisms that alter this external driver to result in a slow freeze-down followed by a rapid melting. Sundby et al. turn to a recent study of theirs dealing with Mn diagenesis to posit that perhaps a similar process occurs with methane, in such a way that it interacts with sedimentary phosphorus diagenesis and then triggers a nutrient-productivity feedback impacting atmospheric carbon dioxide.

First, kudos to the authors for refreshing the shelf-nutrient hypothesis and considering other processes to explain the "still not explained" paleo-productivity records and rapid glacial-interglacial transition. Though conjectural at this point, the basic premise of methane destabilization and changing the diagenetic front in sediments to release phosphate into the overlying water column is tenable at face value at least.

**We thank Dr. Filippelli for the laudatory remarks.**

I have some overall "thought responses" to various aspects of this paper, to be taken as reflections rather than as criticisms of this excellent work.

The authors state beginning on line 62 "The amount of sediment that has to be deposited and eroded on the relevant time scale appears to be far greater than can be supported by observations (Peacock et al., 2006). Another weakness of the hypothesis is the absence of an explicit mechanism that would release phosphate into the aqueous phase once the sediments are eroded."

This is indeed a challenge to the shelf-nutrient hypothesis, particularly if one considers this an on-off switch of glacial-not glacial. But it is not a binary system—as the Earth creeps its way into a glacial period, sea levels successively falls, exposing more and more of the shelf to subaerial exposure and yielding less and less burial space for phosphorus on the shelf. One would expect to see a steady increase in phosphorus delivery to the deep-sea sink, which has a much longer internal residence time wrt to phosphorus given great recycling extent of falling organic matter through a longer water column. Furthermore, once exposed to subaerial weathering by acidic rainfall and by plant colonization, the shelf sediments can be a critical additional source of phosphorus into the deep-sea sink, further increasing phosphorus loading. We interrogated this phenomenon in "Filippelli, G.M., Latimer, J.C., Murray, R.W., and Flores, J.A., 2007. Productivity records from the Southern Ocean and the equatorial Pacific Ocean: Testing the Glacial Shelf-Nutrient Hypothesis. Deep Sea Research II, 54/21-22: 2443-2452," and found that, with a proper lag, one could see the shelf transfer of phosphorus in deep sea sedimentary records.

**We added, on lines 65-67 of the revised manuscript, a sentence in which we highlight this contributing scenario: "Filippelli et al. (2007) proposed that, with a proper lag time, subaerial weathering by acidic rainfall and plant colonization of exposed shelf sediments will release phosphorus to the ocean."**

At face value this finding supports the classic shelf-nutrient model and I think more faithfully documents the real, gradual progression into a glacial interval. It could thus be argued that we need no additional source of phosphorus beyond what is present on the shelfs plus the phosphorus that avoid shelf deposition in the first place in lower sea levels. But as the author of this, there are several weaknesses, or at least unconstrained components, that provide ample room for an additional phosphorus source argument, like the methane one in this Sundby et al. contribution. First is the issue of scale of input. As Sundby et al. point out, there simply doesn't seem to be enough phosphorus transferred from the shelfs to produce the productivity and $CO_2$ response observed in the record. Second is one of timing—how quickly might exposed shelf sediments be colonized by land plants and weathered through acidic processes? Probably not fast enough to get the rapid response seen in the $CO_2$ record.

**We also added the reference to Filippelli et al. (2007) on line 333 of the revised manuscript, where we point out that our scenario helps explain an increase in P flux during the ice age**

Sedimentary methane oxidation and phosphorus diagenesis

Sundby et al. advance a really novel hypothesis that is nevertheless rooted in the stabilization history of methane clathrates. Namely, could the oxidation of hydrates from lowered hydrostatic pressure (lowered sea level) drive a diagenetic front through iron oxyhydroxides, releasing the ample phosphorus stored in this mineral phase? We know that methane hydrates are ample in marine sediments, and that their stability is dictated by pressure and temperature (in marine sediments, it is likely the pressure component that is most at play here). And the oxidate of these hydrates would then release chain of chemical processes that might reduce the overlying iron oxydroxide layer.

Given how much phosphorus is typically trapped in this phase (based on our work and others it is typically 20-30% of the total soluble phosphorus in marine sediments), this seems to be a plausible, and quite significant, additional source of phosphorus to the deep ocean reservoir, and perhaps one that is previously missing in the shelf-nutrient hypothesis model. What I really like about this mechanism proposed by Sundby et al. is its speed—an advancing front of iron oxide reduction would yield "near instantaneous" (on geologic timescales) release of phosphorus into the overlying water column. Plus, this front can progress in stages, as supported by intervals of sea level stabilization observed as glacial intervals progress. This speed and dynamic response is an interesting and welcome addition to the overall concept of phosphorus playing a key role in ocean productivity and biogeochemical cycling in glacial intervals. Of course, as Sundby et al. point out, let's not get too carried away by the power of phosphorus— certainly, you could hit a point where the limiting scale leans toward nitrogen, and it is important to keep the N/P ratio in mind when reconstructing nutrient limitation for various ecosystems.

**We thank the reviewer for appreciating our scenario. We agree with him that the P fluxes generated by the processes we discuss may be significant, since the reactive P sequestered with iron oxides represents a significant fraction of the oceanic pool, as discussed in Section 3.1, and the quantitative estimates presented in Section 3.7 indeed indicate that this P may be equivalent to the current dissolved phosphate inventory in the ocean. As mentioned by the reviewer, we emphasized in the original manuscript (line 254) that destabilization of methane hydrates and P-release to the overlying waters will occur gradually or episodically as sea level lowers gradually or episodically through the glacial period.**

**We do not rule out the idea that N availability and limitation is involved in the complex history of oceanic productivity, and we point this out in section 4.1.**

**RC2**: David Archer

This is a valuable and well-written paper clearly worthy of publication. The new idea is to combine changes in methane hydrate stability driven by sea level change to diagenesis chemistry that would release phosphorus to the ocean, stimulating biological production and ultimately possibly the enigmatic decrease in the $CO_2$ concentration of the atmosphere. Any new mechanism for volatility of the ocean phosphate reservoir is interesting, because of the profound biological impact of the P in the ocean.

**We wish to thank Dr. Archer for his positive and incisive comments.**

I think it would be illuminating to put characteristic depth and concentration scales into Figure 2, to try to constrain how much phosphorus could be mobilized on which time scales. On the shallow end of the depth range, the hydrate stability zone is thin, so the released P would be near the sea floor, where it could diffuse to the ocean relatively quickly. On 100,000 years, the mean free path for molecular diffusion is about 50 meters. But if the stability zone is 500 meters thick, a change at the bottom of the stability zone would be pretty isolated from the ocean. In this way perhaps one could estimate the amount of methane that is properly poised to release phosphorus on a glacial time scale, which would tell you also the magnitude of the potential P dose. Putting characteristic depth and concentration units onto the profiles in Figure 2 would allow you to calculate a diffusive gradient and flux for P, another way to address the same issue, I guess. My hunch is that this depth-restriction of where this process could be relevant on 100,000-year time scales might diminish the potential scope for driving ocean P cycles, but that would not diminish for me at all the value of this paper or its worthiness of publication.

**A very good point. In response, we added characteristic depth and concentration scales to Figure 2 and expanded the figure caption. We only hinted at these in section 3.7 of the original manuscript and they were implicit but unspecified in Figure 2. They are, indeed, critical factors in determining the responsiveness of the sediment to sea-level change.**

**Dr. Archer indicates that if the stability zone is 500 meters thick, a change at the bottom of the stability zone would be pretty isolated from the ocean. We agree with him: little of the SRP released 500 m below the seafloor would reach the SWI. Nevertheless, as mentioned in our original manuscript (section 3.4, line 220-225) Borowski et al. (1999) and Gorman et al. (2002) show that methane gas, produced by hydrate dissociation at the lower stability boundary, can be injected well into the overlying hydrate stability zone. The data indicate that methane, in the form of free gas, can migrate hundreds of meters through the hydrate stability zone before re-forming as solid phase hydrate.**

**We made the following addition in section 3.7. on lines 317-328 of the revised manuscript: "The potential flux of phosphate fed by the ascent of the SMT is significant if the diffusional transport of P is sufficient for this phosphate to reach the sediment surface. The flux equation for transport via molecular diffusion is:**

$$J = -\varphi D_s \, (\Delta(SRP)/\Delta z)$$

**where J is the phosphate flux across the sediment-water interface; $\varphi$ is the mean porosity of the sediment; z is the depth coordinate; $(\Delta(SRP)/\Delta z)$ is the phosphate concentration gradient. Ds is the bulk sediment molecular diffusion coefficient, assumed to be equal to $D_s = D_o/(1-\ln \varphi^2)$ (Boudreau, 1996), where Do is the diffusion coefficient in water at in situ temperature. Considering a porosity of 0.7 and Do of 0.0124 $m^2$ $yr^{-1}$ for $HPO_4^{2-}$ at 5°C (Schulz and Zabel, 2006), a mean flux of $1\times10^{10}$ mol $yr^{-1}$ from a sediment surface of 23 × $10^6$ $km^2$ (=$4.35\times10^{-4}$ mol $m^{-2}$ $yr^{-1}$) requires a concentration gradient $(\Delta(PO4)/\Delta z) = J/(- \varphi D_s)$ of 0.086 mol $m^{-3}$ $m^{-1}$ or 86 µmol $L^{-1}$ $m^{-1}$. Such a gradient is only 2 to 4 times higher than the highest gradient observed on continental slope sediments by Niewohner et al. (1998) or Charbonnier et al. (2019), suggesting that molecular diffusion is not an obstacle to the transfer of the inventory of mobilizable phosphate toward the sediment-water interface, as long as the concentration gradient becomes high, as our scenario predicts."**

I think it would be worth mentioning, in the background to the ocean nutrient inventory as a driver of atmospheric $CO_2$, that the extent of this effect should be reflected in the plankton – benthic $\delta^{13}C$ difference, a measure of the strength of the biological pump. I think this "$\Delta\delta13C$" says that the biological pump is not the entire answer to the glacial $CO_2$ drawdown. That also does not diminish the value of this paper at all; the proposed coupling between methane and phosphorus cycles should clearly be elucidated, as this paper nicely does.

**Cartapanis et al. (2016) provide evidence of greater OM burial during the glacial period, but do not include isotopic data to confirm the role of the biological pump. This reference appears in Table 2 of the original manuscript. Nevertheless, we added a sentence on lines 54-55 "This scenario is supported by reports that global rate of organic matter burial is maximal during the glaciation (Cartapanis et al., 2016; Boyle, 1986; Wallmann, 2010)."**

Line 105: A citation for the "until recently" would be useful. I note that there is further discussion of this topic later in the paper.

**We added the reference to Egger et al. (2015) to support this statement.**

Line 127 My distant recollection is that there is a lot of particulate phosphorus in rivers, stuff that gets scavenged in the estuary double-layer-collapse sedimentation zone. It might be helpful to clarify how this flux fits into the P budget. Could it be that it is this phosphorus flux that gets periodically mobilized, maybe along with biologically-deposited phosphorus? Thinking about this might be complicated by estuaries that exist or not through glacial cycles, possibly changing where the P deposits. Perhaps that would increase the potential P mobilization, if you're

not limited by P that was deposited by primary productivity during the past glacial cycle (which is a very clever idea).

**Many estuaries serve as nutrient traps and these P-rich estuarine sediments are likely to be remobilized (or colonized by plants (Filippelli et al., 2007)) when sea level decreases and exposes shelves, but, like the shelf-nutrient hypothesis, a mechanism to release SRP from these sediments would be required.**

Line 277  I'm not sure I agree that a linear P gradient implies that there is no impediment of diffusion by adsorption onto solid phases. There is clearly a diffusive flux, but it could be linear in a long-term steady state, with a higher concentration of adsorption P going with higher porewater concentrations at depth; the pore waters could still be buffered by the sediment. That might be important for thinking about the time dependence of the phosphate concentration and hence cycles in the diffusive flux. In a more buffered case, I guess the relevant depth range for driving P cycles would get shallower.

**There might be some authigenic precipitation, but we believe (Sundby et al., 1992) that, given that P adsorption is an equilibrium process and, barring significant co-precipitation (or solid-state diffusion) with iron oxides, iron oxide surfaces should be mostly saturated.**

**Sundby B., Gobeil C., Silverberg N. and Mucci A. (1992) The phosphorus cycle in coastal marine sediments. Limnol. Oceanogr. 37: 1129-1145.**

**As explained in our response to the comment about the diffusive flux (see above), molecular diffusion is not an obstacle to the transfer of the inventory of mobilizable phosphate toward the sediment-water interface, as long as the concentration gradient becomes high enough.**

RC3: Sabine Kasten

The „Ideas and Perspectives" manuscript by Sundby et al. presents a new potential scenario that adds to the suite of processes that may have contributed to the rapid climate changes observed along glacial/interglacial transitions - namely the documented rapid increase in atmospheric $CO_2$ concentrations during deglacials or more precisely prior to glacial terminations. The authors suggest a mechanism that links sealevel, rates of anaerobic oxidation of methane and benthic fluxes of phosphate in this way contributing a new mechanism to the so-called „shelf nutrient hypothesis" initially developed by Broecker (1982).

I find it very interesting and scientifically fruitful to put forward this scenario and in particular in this way stimulate discussion over a broad spectrum of disciplines.

Being a „sediment diagenesis" person myself, I hope you understand that I will mostly focus on related issues.

**We thank Dr. Kasten for her constructive comments. The four co-authors, being seasoned sediment diageneticists, provide informed responses to the various comments.**

The authors propose the following suite of conditions and processes: 1) during glacial sea level lowstands hydrostatic pressure is lowered, 2) this enhances the upward flux of methane from underlying gas hydrates and shifts the sulfate/methane transition (SMT) to a shallower depth in the sediment, 3) as a consequence rates of anaerobic oxidation of methane (AOM), hydrogen sulfide formation, reductive dissolution of reactive Fe oxide minerals by sulfide and associated release of adsorbed phosphate into the pore water all increase, 4) this induces higher upward (and downward) fluxes of phosphate, and 5) also increases the flux of phosphate across the sediment/water interface into the bottom water.

While I fully agree with the conditions and processes in points 1 to 4, I am not convinced with the statement given as point 5 above – namely elevated benthic fluxes of phosphate into the water column during a shallower position of the SMT during glacials. Unfortunately the authors have neither presented nor discussed data that demonstrate that phosphate fluxes into the bottow water are indeed higher when the SMT is positioned at shallow sediment depth.

**We clarified this point in the revised version of the manuscript on line 211 and thereafter. First, we point out that it is not the difference in the vertical location of the SMT in the sediment column at two different sites that determines the phosphate flux, but rather the migration of the SMT closer to the sediment-water interface during the glacial cycle that increases the SRP concentration gradient and the flux.**

**Our manuscript is listed under "ideas and perspectives" precisely because there are no new data available to present or to discuss. We therefore propose a scenario for the diagenetic remobilization of inorganic phosphate that is based on reasonable assumptions relative to what is understood about sediment diagenesis. We exploit the idea that diagenetic processes in the sediment column can be driven by an electron flux from below (in this case, a methane flux) and that this flux can vary with changing physical conditions (sea level/hydrostatic pressure, temperature) during the glacial cycle. We explore the impact this may have on the flux of inorganic phosphate from the sediment to the ocean; we are testing a hypothesis, not discussing data.**

**Dr. Kasten**: They exclusively discuss gravity core data, however, completely neglect the part of a sediment that is key in determining the flux of phosphate across the sediment/water interface – namely the sediment surface. Benthic fluxes can only be assessed if the pore-water gradients of phosphate in the uppermost few decimeters of the sediments are determined. In order to quantify the diffusive flux of phosphate from the sediment back to the bottom/deep water – and thus into the oceanic reservoir - pore-water data of multiple cores or push cores (or benthic chambers) are required, which allow a proper disturbance-free sampling of the sediment surface. Hence, also the schematic representation given in Fig. 2 is not correct, at least as given. With the constant concentrations of phosphate displayed in the uppermost part of the graphs, there is/can be no phosphate flux across the sediment-water interface (SWI) – neither during interglacials nor glacials.

**We have no data about the actual benthic fluxes of soluble reactive phosphate (SRP) during glacial periods. Therefore, we turn our attention to our diagenetic scenario and propose that destabilization of methane-hydrates upon sea-level fall (lowers the overlying hydrostatic pressure) allows the SMT to rise closer to the sediment-water interface. This solubilizes large amounts of the inorganic phosphate that was initially adsorbed to or co-precipitated with iron oxides.**

**Under our scenario, the displacement of the SMT generates a strong concentration gradient and drives a flux of SRP towards the sediment-water interface. Whereas some of the SRP may be sequestered by authigenic iron oxides near the SWI, SRP will nevertheless diffuse out of the sediment, as it is known to do under most circumstances. As a result of organic matter remineralization, pore-water SRP concentrations immediately below the SWI nearly always exceed the overlying water concentrations, even though the former are buffered by adsorption to detrital and authigenic iron oxides in the oxic sediment, and thus support a SRP flux out of the sediment, a phenomenon that has been confirmed by a wealth of direct flux measurements in freshwater and marine environments.**

**Given the added SRP flux generated by the vertically migrating SMT during the glacial period, the concentration of maximum buffering capacity of the sediment (or zero equilibrium phosphate concentration (EPC$_0$), see Sundby et al. (1992) for details, a concept first introduced by Froelich (1988)), will rapidly be exceeded and the pore-water SRP concentration gradient across the SWI will strengthen. If the flux of SRP from below is large, the buffering (or adsorption) capacity of the authigenic iron oxides may be overwhelmed and the linear gradient could extend all the way to the SWI. It is difficult to imagine that the entire SRP flux that is generated by the vertical migration of the SMT could be completely trapped below the sediment water interface in a layer that would be constantly supplied with and enriched in inorganic phosphate. We are convinced that a large fraction of the SRP generated in this way will diffuse to the overlying water column, a more reasonable hypothesis. Accordingly, we have modified the fourth panel of Figure 2 to illustrate that the iron oxide-buffered pore-water SRP concentration (a vertical line) immediately below the SWI increases in response to the increased flux of SRP from below, although the SRP concentration gradient may extend all the way to the SWI if the SRP flux is large enough to saturate all adsorption surfaces. A detailed explanation**

**was also added to the figure caption, and the following sentences were added on lines 328-331: ". A mean diffusive SRP-flux of $4.35 \times 10^{-4}$ mol m$^{-2}$ yr$^{-1}$ is lower than the phosphate return flux fuelled by organic matter regeneration at the sediment-water interface in continental margins (Hensen et al., 1998; Colman and Holland, 2000). However, the increased flux of SRP from below could increase the pore-water concentration gradient immediately below the sediment-water interface (see Fig. 2 for explanation)."**

Froelich, P. N. (1988). Kinetic control of dissolved phosphate in natural rivers and estuaries: A primer on the phosphate buffer mechanism. Limnology and Oceanography, 33: 649-668.

Sundby B., Gobeil C., Silverberg N. & Mucci A. (1992). The phosphorus cycle in coastal marine sediments. Limnology and Oceanography, 37, 1129-1145.

There is also some contradiction and imprecise discussion with respect to the assumed phospate concentrations in the oceanic reservoir/deep waters and the major particulate carrier phases that transport phosphorus into the sediment (organic matter, Fe-bound phosphate) throughout the manuscript, which needs to be sharpened and better structured. Moreover, parts of the manuscript contain several incorrect statements and assumptions. concerning the stability of methane gas hydrates and the characterisitics of methane transport both is dissolved and gaseous form in marine sedimentary environments.

**We respond to these comments below, in the section on specific comments.**

I was also wondering whether considering issues related to the (potential) release of methane during glacial/interglacial sea level changes really falls into the scope of this manuscript.

**We mention this early in the text, which seems pertinent, since in our scenario, the diagenetic processes that generate the enhanced inorganic phosphate flux are fueled by this methane.**

The scenario presented also does not consider the processes that occur in exposed shelf sediments during glacial sea level low-stands. As presented and suggested by Kölling et al. (2019) aeration/oxidation of shelf sediments during sea-level low stands will oxidize reduced iron sulfide minerals resulting in the formation of abundant reactive Fe(III) that will certainly serve as an efficient trap for phosphate either by co-precipitation or adsorption of phospate on Fe(III) mineral surfaces. I would therefore assume, that these (bio)geochemical processes acting in exposed shelf surface sediments are a significant sink for phosphate during glacial periods. How important is this potential sink compared to the (potential) release of phosphate from continental margin sediments into the oceanic reservoir?

**We thank Dr. Kasten for bringing this article to our attention. The article by Kölling et al. (2019) proposes a mechanism of CO$_2$ release in response to the oxidation of pyrite to iron oxides during the emersion of continental shelf sediments when the sea level drops during the glacial period. This article does not refer to the impact of this mechanism on the phosphate pool. Since this scenario refers to the formation of iron oxides in areas that are above sea level during glacial periods, these oxides cannot trap marine phosphate unless eroded and resuspended in the open ocean. It is therefore impossible to compare the phosphate fluxes resulting from such a scenario with our own. The proposed mechanism could contribute to the rise in atmospheric CO$_2$ during the glacial period and, as such, is now highlighted in the revised manuscript when we discuss oceanic processes leading to an increase in atmospheric CO$_2$ (lines 393-394).**

So, to conclude, the major issue I have is that the authors have not convincingly shown and discussed that benthic PO4 fluxes from continental margin sediments were/are indeed higher during glacial times or in general during times of a shallower position of the SMT – mainly because they have not at all considered and discussed pore-water data for the sediment surface proper. I think that such a scenario could be easily and convincingly tested by comparing sites – i.e. data of MUC or push cores that allow to sample the sediment surface proper - where the SMT is located at different sediments depths. Perhaps also pore-water phosphate data for active seep sites are available/published. In this way the authors could test their hypothesis that during periods of a shallower position of the SMT (and/or periods of active methane seepage) phosphate fluxes across the SWI are/were indeed higher than with a deeper SMT.

**To help validate our hypothesis, Dr. Kasten suggests that we compare dissolved phosphate profiles in pore waters near the SWI of sediments from different sites where SMT is currently at different depths. The suggested exercise would be useless since these are steady-state conditions, unlike our proposed transient-state scenario in which the SMT migrates vertically in response to a change in sea level, destabilization of methane hydrates, and a consequent upward flux of dissolved methane, so that the resulting sulfide front will reductively dissolve iron oxides that accumulated above it and release the inorganic phosphate associated with them. In summary, as noted above and emphasized in the revised manuscript (lines 211-215), it is not the depth of the SMT that defines the phosphate fluxes in our scenario, it is the change in the depth of the SMT that generates an increase in the SRP flux.**

I also think that the argumentation can be significantly strengthened by better structuring the manuscript with respect to 1) precisely stating the postulated changes in phosphate concentrations in the oceanic reservoir during glacial and interglacial times, 2) precisely and consistently discussing the main carrier phases of P into marine sediments – namely organic matter and Fe-bound P, and 3) also carefully checking the parts of the manuscipt where you discuss (changes in) hydrate stability and the processes transporting methane through sediments and across the SWI.

Specific comments and corrections

Line 13: „additional" to what precisely?

**The use of "additional" is justified as the potential flux of SRP generated by the destabilization of methane hydrate and consequent vertical migration of the SMT is above and beyond the steady-state flux of SRP sustained by organic-P remineralization during early diagenesis.**

Ls. 15/15: Here you only mention biomass as a carrier phase to return/carry P back to the sediment. What about Fe-bound P, which – as you state on page 3 – is the key issue of your study/scenario?

**In this statement, our intent was to note that increasing the phosphate flux to the oceanic reservoir should stimulate the biological pump.**

Ls. 20/21: This somehow contradicts your statement in lines 13-15 that phosphate fluxes were/are higher during glacial. Why does the deep water then has lower P concentrations during/at the beginning of a deglaciation? I find this very confusing.

**The sentence on lines 20/21 refers to the beginning of the deglaciation when the phosphate released previously during the glacial period has been consumed, whereas the sentence on lines 13/15 refers to the period of sea-level fall.**

Page 2: In this context, I would like to draw your attention to the recent paper by Kölling et al. (2019) who have presented a scenario explaining the rapid increase in atmospheric $CO_2$ concentrations prior to glacial terminations – also linking sealevel changes to atmosphereic $CO_2$ concentrations, more precisely considering high rates of pyrite oxidation on continental shelves exposed during glacial sealevel low-stands.

Kölling, M, Bouimetarhan, I, Bowles, MW, Felis, T, Goldhammer, T, Hinrichs, K-U, Schulz, M, Zabel, M (2019): Consistent $CO_2$ release by pyrite oxidation on continental shelves prior to glacial terminations.- nature geoscience **12**, 929-934. doi 10.1038/s41561-019-0465-9.

**We thank Dr. Kasten for bringing this article to our attention. The reference was added to line 42 of the revised manuscript.**

1. 46: methane hydrates

**The correction was applied, as proposed.**

Ls. 46 ff: I do not agree with the scenario that sea level fall-induced lowering of the hydrostatic pressure necessarily leads to a transport of significant amounts of methane into the water column and subsequently into the atmosphere. Of course a lower sea level will induce a thinning of the gas hydrate stability zone (GHSZ). If methane is transported by diffusion, the methane transported upward from gas hydrates will be more or less completely oxidized/consumed before it can reach the water column. If methane is transported in the form of free gas, these bubbles can reach the lower water column. However, numerous studies have shown that they do not significantly contribute to transport of methane into the atmosphere because methane is rather oxidized areobically in the water column and/or dispersed by horizontal advection and dilution. Moreover, rising bubbles constantly change their internal gas composition due to counterdirected diffusion of gases (methane out, N and CO2 in) across the bubble/water interface (cf. studies by Mc Ginnis or Leifer). As a consequence even in shelf settings with shallow water depth, methane does not reach the atmosphere in considerable amounts (e.g. Mau et al., 2015; Biogeosciences; Geprägs et al., 2016; G3).

Ls. 82 ff.: I also do not fully understand why you discuss the (very controversial) role of methane in the atmospheric carbon cycle over glacial/interglacial changes – please see comments above. Is this really in the scope of your manuscript?!

**The sentences on lines 46-50 of the original manuscript simply reiterate what previous authors have proposed and end with a statement, based on conclusions from more current literature, that the scenario is questionable because the methane would likely be oxidized before reaching the overlying water column. Hence, the text is appropriate as it provides a chronological, literature review of the potential link between methane, sea-level fall and global warming.**

**We fully agree with Dr. Kasten as the sentence on line 82-84 ("We will also re-examine the idea proposed by Paull et al. (1991, 2002) that sedimentary methane may be directly involved in the atmospheric carbon cycle by escaping to the atmosphere during periods of sea-level fall.") was inadvertently left in the manuscript from a previous version and this may have led to some confusion. It was not our intention to describe all the processes that prevent methane accumulating in marine sediments from reaching the atmosphere. We simply wanted to point out that the scientific community has long been interested in the link between methane hydrate stability and glacial/interglacial fluctuations. The sentence was removed from the revised manuscript. Dr. Kasten rightly writes that most of the methane is oxidized in the pore waters before reaching the ocean water column. In fact, in the original manuscript, we wrote: "…but it is doubtful that the vast quantities of methane that would be required to trigger substantial warming could reach the atmosphere before being oxidized in the sediments and the overlying water (Archer et al., 2000; Archer, 2007).". We added one of the references recommended by the reviewer to the revised manuscript (line 191). It is precisely the process of anaerobic $CH_4$ oxidation that is the basis of our scenario, where the oxidation of methane results in an electron transfer that allows the reduction of iron oxides and the release of large quantities of inorganic phosphate to the pore waters. This scenario is laid out in detail in section 3.3 of the original manuscript.**

1. 116: … downward (and upward) diffusion …; into sulfate- and sulfide-depleted pore water below the SMT ….

**This is an interesting observation: the data show both upward and downward transport of phosphate within the same core (März et al., 2008). The suggested explanation for this is that vivianite forms (authigenesis) deep in the sediment.**

1. 119: drive instead of imply:

**Replaced by "drive', as requested**

Ls. 124 ff.: Please, also check the earlier papers by Hensen et al. in this context:

C Hensen, H Landenberger, M Zabel, HD Schulz (1998) Quantification of diffusive benthic fluxes of nitrate, phosphate, and silicate in the southern Atlantic Ocean. Global Biogeochemical Cycles 12 (1), 193-210.

C Hensen, M Zabel, HN Schulz (2006) Benthic cycling of oxygen, nitrogen and phosphorus. Marine Geochemistry, 207-240.

**The article by Hensen et al (1998) presents the analysis of an impressive dataset. The global benthic flux estimate of Hensen et al. (1998) ($3.2 \times 10^{11}$ mol/yr) is significantly lower than that proposed by Colman and Holland (2000) ($12.5 \times 10^{11}$ mol/yr). The flux measurements of Hensen et al. (1998), however, apply mainly to deep marine environments (>1000 m) whereas those of Colman and Holland (2000) focus more on continental margins. Qualitatively, the conclusions of Hensen et al. (1998) agree with those of Colman and Holland (2000): benthic fluxes greatly exceed river fluxes. A sentence was added the revised manuscript (lines 130-131) that highlights the results of Hensen et al., (1998) in the Southern Atlantic Ocean: "Hensen et al. (1998) reported phosphate fluxes that are slightly lower but of the same order of magnitude from deep (> 1000m) sediments of the Southern Atlantic Ocean."**

Ls. 130 ff.: Can you specify? In which kind of environment? Oxygen minimum zones? In this context please also consider the release of Fe-bound P in sediments underlying continental margin oxygen minimum zones – as also typically observed in anoxic lakes (sometimes referred to as internal fertilization)

**Phosphate is released from continental margin sediment to the overlying seawater. One example is described in Sundby et al. (1992) where SRP fluxes to the overlying water were estimated on 27 undisturbed box cores collected in the Laurentian Trough in Canada. Other examples are given by Colman and Holland (2000). They examined pore-water profiles of soluble reactive phosphorus on nearly 200 sediment cores (some of which were collected by divers). They concluded that the return flux of phosphate from continental margin sediments to the overlying water was more than one order of magnitude larger than the riverine flux of total dissolved phosphorus to the ocean. "Hensen et al. (1998) reported phosphate fluxes that are slightly lower but of the same order of magnitude from deep (> 1000m) sediments of the Southern Atlantic Ocean."**

Ls. 144 ff.: As already stated above, the sediment surface (upper sediment column) is not only important with respect to bioirrigation and bioturbation, but the key locus that determines the diffusive flux of phosphate across the sediment surface into the bottom water.

**We agree that the sediment surface (upper sediment column) is important with respect to bioturbation and bioirrigation and as the locus that determines the diffusive flux across the sediment surface into the bottom water. We modified the penultimate sentence of this paragraph of the revised text (line 148) to further emphasize this point: "Soluble forms of phosphorus are transported by diffusion along concentration gradients that often develop in sediment pore waters and across the sediment-water interface. Transport via bioirrigation can also be important in the upper sediment column."**

Ls. 166 ff.: No, $Fe^{2+}$ generally does not diffuse across the Fe(II)/ Fe(III) redox boundary but is mostly oxidized at this redox boundary by nitrate and generally does not make it further up to the lower boundary of the oxic zone (cf., Froelich et al., 1979; Berner 1981; Kasten et al., 2003).

**If there is a concentration gradient of Fe(II) across the Fe(II)/Fe(III) boundary, then this gradient can drive a flux. The magnitude of the flux will depend on the strength of the gradient, which depends on the rate of reoxidation. The use of "transport across the Fe(II)/Fe(III) boundary …" is perhaps what triggered this comment. It is understood that there must be a concentration gradient across the Fe(II)/Fe(III) boundary to drive a flux, but we agree that much of the Fe(II) will be oxidize just above this boundary and, thus, we will substitute "across" by "to". Keep in mind that a similar process takes place with Mn(II) but, because of it slow oxidation kinetics, Mn(II) can diffuse well above the Mn(II)/Mn(IV) boundary before being oxidized and precipitated as authigenic manganic oxides.**

Ls. 174 ff.: No, the reactivity of sedimentary Fe oxides does not necessarily decrease with time or depth of burial. As numerous studies have shown for shelf and continental margin sediments (Riedinger et al., 2005, GCA; Riedinger et al., 2017, Frontiers in Earth Sciences; März et al., 2008; Oni et al., 2015, Frontiers in Microbiology; Köster et al., 2021, G3) high amounts of reactive Fe (III) minerals can be buried to substantial sediment depth if sedimentation rates are high (as typical for continental margin settings) - thus limiting „sulfide exposure time".

**We agree that the reactivity of sedimentary iron oxides does not necessarily decrease with time. Accordingly, we modified the sentence on line 174 to: "Once buried, authigenic iron oxides may undergo an aging process … ".**

Ls.: 200 ff.: The depth of the SMT may not only fluctuate over glacial/interglacial timescales (e.g. Henkel et al., 2012, GCA) but also be affected by increases/changes in the upward flux of methane induced by overpressuring of the underlying gas reservoir and/or triggered by earth quakes or sediment mass movements (e.g., Fischer et al., 2013, Nat. Geoscience; Henkel et al., 2011, G3; Henkel et al., 2012, Springer book on Submarine Mass Movements).

**We agree that the depth of the SMT may also fluctuate and the upward methane flux may increase in response to over-pressuring of the underlying gas reservoir, earthquakes and sediment mass movements. The reference to Henkel et al. (2012) was added to line 203 and the following sentence was added to lines 208-210 of the revised manuscript: "The depth of the SMT may also fluctuate and the upward flux of methane increase in response to over-pressuring of the underlying gas reservoir, earthquakes or sediment mass movements (Henkel et al., 2011; Fischer et al., 2013)".**

Ls. 206 ff.: I do not agree with this statement. A shallow SMT does not necessarily mean that the flux of phosphate across the SWI is increased. Please, give examples/data of MUC or push cores that demonstrate this.

What do you mean with „instantaneous" flux? This is not clear to me at all.

**We agree. The flux of phosphate across the sediment-water interface is determined by the strength of the concentration gradient, but the gradient will increase if the SMT suddenly migrates upwards and Fe oxides that accumulated in the sediment above it are reductively dissolved by sulfides (a by-product of AMO), releasing the associated phosphate to the pore water. Accordingly, the sentence on lines 212-213 was modified to: "The closer the SMT migrates to the sediment surface, the greater is the flux of sulfate into the sediment and the shorter is the path that phosphate travels before it escapes the sediment.".**

**By 'instantaneous flux', we mean the flux at any given moment.**

Ls. 119 ff.: This paragraph on gas hydrate stability and transport of gas containes several flaws and imprecise statements. L. 119: methane is not „produced" by hydrate dissociation but released from the hydrate phase.

The upper boundary of the gas hydrate stability zone (GHSZ) is determined by temperature and water depth/pressure and not by sediment accumulation.

What do you mean with „instantaneous" methane flux? Bubble ebullition, migration of free gas? Please, specify.

**The reviewer likely meant Ls. 220. We agree that methane is not produced by hydrate but released upon the destabilization/dissociation of the hydrate phase. Hence, "produced" was be replaced by 'released' on line 226 of the revised manuscript. We agree that the gas hydrate stability zone (GHSZ) is determined by temperature and water depth/pressure, but as sediments accumulate the overburden increases and the GHSZ migrates with it.**

**Again, by 'instantaneous flux', we mean the flux at any given moment.**

Ls. 238-240: Again, what do you mean with „instantaneous" methane flux? I do not at all agree with the statement in this sentence. At least you should give examples and also precisely state which part/interval of the sediment you refer to. The fluxes of both constituents can indeed by higher in the deeper sediments around the SMT but not necessarily at/across the SWI.

**By 'instantaneous flux', we mean the flux at any given moment.**

Ls. 244-245: No, this is not exactly what we see. There is both a downward flux and an upward flux of Fe2+ towards the SMT (cf., Riedinger et al., 2005, GCA, 2017, Frontiers; März et al., 2008).

**There is no contradiction with having both upward and downward fluxes within a sediment layer. Riedinger et al. (2014) observed both an upward and a downward directed flux of Fe(II). The direction of the flux and the magnitude of the flux depend on the concentration gradient of the species that are diffusing.**

1. 249: No, as already stated above, I do not agree that during a shallow location of the SMT the flux of phosphate across the SWI is increased (at least I have seen no data).

**Please see our response to the comment on Ls. 206 : The location of the SMT in the sediment column is critical since this is where reduction of iron oxides releases phosphate to the pore water. The closer the SMT migrates to the sediment surface, the greater is the flux of sulfate into the sediment and the shorter is the path that phosphate travels before it escapes the sediment (see lines 212-213 of the revised manuscript).**

Ls. 254 ff.: The benthic phosphate flux is also significantly dependent on the redox/oxygen conditions of he overlying bottom water. Cf. comment for l. 130.

**We agree. This was clearly demonstrated in sediment incubation experiments carried our by Sundby et al. (1986).**

**Sundby, B., Anderson, L., Hall, P., Iverfeldt, A., Rutgers van der Loeff, M. and Westerlund, S.: The effect of oxygen on release and uptake of iron, manganese, cobalt, and phosphate at the sediment-water interface. Geochim. Cosmochim. Acta, 50, 1281-1288, 1986.**

Ls. 265 ff.: I do not fully agree with the discussion in this paragraph. It may be that Fe(III) phases arriving at the seafloor are already close to saturation with respect to potential sorption sites for phosphate. This does, however, not hold true for the freshly formed/authigenic Fe oxides at the Fe redox boundary. You have also discussed this in previous paragraphs of the manuscript. Numerous data show that most of the upward diffusing phosphate is trapped in the vicinity of the Fe redox boundary, i.e. the pore-water gradient of phosphate changes and only minor amounts of phosphate make it to the overlying bottom water.

**We agree that numerous data show that much of the upward diffusing phosphate in sediment pore water is trapped in the vicinity of the Fe redox boundary. On the other hand, the case has been made that large amounts of dissolved inorganic phosphate actually escape the continental margin and deep-sea sediments into the water column (see Hensen et al., 1998; Colman and Holland, 2000). A similar claim is supported by nearly 30 box core-based phosphate pore-water profiles in the Gulf of St. Lawrence using state-of-the-art sampling and millimeter-scale pore water extraction techniques (Sundby et al., 1992).**

**Whereas some of the SRP may be sequestered by authigenic iron oxides near the SWI, SRP can nevertheless diffuse out of the sediment. This can be observed under nearly all the sedimentary conditions that result from organic matter remineralization: pore-water SRP immediately below the SWI nearly always exceed the overlying water concentrations and, thus, SRP diffuses out of the sediment. If the flux of SRP from below is very large, the concentration of maximum buffering capacity of the sediment (or zero equilibrium phosphate concentration (EPC$_0$), see Sundby et al. (1992), a concept first introduced by Froelich (1988)) will be rapidly**

**exceeded and the pore-water SRP concentration as well as its concentration gradient across the SWI will build up. Ultimately, the buffering (or adsorption) capacity of the authigenic iron oxides might be overwhelmed and the linear gradient could extend all the way to the SWI.**

Ls. 270 ff.: You are absolutely right that phosphate may diffusively migrate in pore water over relatively large distances of several meters and more – i.e. Niewöhner et al. (1998). This is particularly true for sediments underlying high productivity areas like off Namibia, in which only low amounts of reactive Fe(III) are preserved at depth due to the high rates of sulfate reduction/sulfide production. However, this is certainly not the typical situation in the vast area of continental margin/slope depositional settings. Moreover, please consider that this Niewöhner et al. (1998) paper (and others you have discussed in your manuscript) only shows data for gravity cores. During sediment sampling with gravity cores the uppermost decimeters of the sediments are always lost, so these data do not allow to assess the flux of phosphate across the SWI.

**We found several examples in the literature where the SRP shows a regular gradient over several meters in long cores. Charbonnier et al. (2019) reported such profiles from cores taken on the Bay of Biscay continental slope, which is not an area of exceptionally high productivity.**

**We agree that there are problems with using gravity cores to assess the flux across the SWI. On the other hand, examples exist where other types of cores that preserve the integrity of the SWI have been used. We refer the reviewer to the box core collection from the Gulf of St. Lawrence (Sundby et al., 1992.)**

**Charbonnier, C., Mouret, A., Howa, H., Schmidt, S., Gillet, H., and Anschutz P.: Quantification of diagenetic transformation of continental margin sediments at the Holocene time scale, Cont. Shelf Res., 180, 63-74, 2019.**

Chapters 3.7 and 4, pages 9 ff.: I do not agree to several of the assumptions presented and discussed here. First, I find it confusing that in your calculation of the sedimentary inventory of P you do not consider organic matter (OM) – although you highlight the OM burial pathway as a key carrier phase of P to the sediment in the abstract and other parts of the manuscript. How much is it compared to the Fe-bound P and how much P is released to the pore water as a consequence of mineralization of OM (compared to reductlive dissolution of Fe(III) minerlas by sulfide? Second, I do not agree that the calculated inventory of Fe-bound phosphate has a chance of ultimately ending up in the water column.

**Our scenario involves the Fe-bound phosphorus. Dr. Kasten signals her disagreement with our calculated inventory of phosphorus and with our assertion that the calculated inventory ultimately ends up in the water column. Given the absence of a solid-phase phosphorus-inventory in the literature, we had no choice but to make our own estimate of the Fe-bound phosphorus inventory in the sedimentary reservoir. What is of particular interest is that the estimated inventory of solid phase phosphorus in marine sediment is of the same order of magnitude as the oceanic reservoir, i.e. the oceanic inventory. This supports our two-box model scenario until other estimates of the sedimentary inventory become available.**

**In the revised version of the manuscript, we wrote on line 308: "Therefore, 100,000 years of sediment accumulation corresponds to a sedimentary column that is 15 to 42 m thick, containing between $0.69 \times 10^{15}$ and $1.93 \times 10^{15}$ moles of mobilizable Fe-bound phosphorus. "**

You also have not discussed whether you think that methane transport occurs via diffusion and/or advection – i.e by methane seepage/bubble ebullition. If methane transport mostly occurs via diffusion then both methane and phosphate – although initially released into pore water - will be mostly trapped in the sediments overlying the gas hydrates (methane at the SMT) and phosphate at the Fe redox boundary close to the sediment surface.

**Within a simple scenario such as ours, one has to overlook some of the details such as the mode of transport (for example methane bubbles vs. seepage). The subject is important, as witness the use of acoustics to locate the lower boundary of the SMT. For a brief discussion, see the following two paragraphs.**

From recent studies in continental margin oxygen minimum zones we see that phosphate is only transported from the sediments into the overlying bottom water at high rates under conditions of oxygen-depleted/anoxic bottom waters or at times of active methane seepage – i.e. ebullition of gaseous methane. – phosphate may be transported at elevated rates into the overlying water column. – in a process similar to mixing of pore water into the bottom water produced by bioturbating/bioirrigating benthic organisms.

If methane transport occurs in the gaseous form – i.e. as bubble ebullition, this occurs along preferential migration pathways, which are spatially (and temporarily) restricted and thus AOM occurring close to theses sites/pathways of gas migration also does not have the capacity to drive reductive Fe(III) reduction over a broad front. Therefore certainly not being able to reductively mobilize the calculated Fe-bound P inventory.

Page 11, upper paragraph: Here, you only speak of OM as a carrier phase to transfer P to the sedimentary reservoir. See also previous comment above.

**Biomass is an important carrier phase, as is phosphate adsorbed to inorganic phases.**

Chapter 4.3: Here are numerous flaws and imprecise statements with respect to the GHSZ and the transport of methane in marine sediments. A few examples:

Ls. 349/350: No, during sea level drop the upper boundary of the GHSZ moves down (not up). Thus the GHSZ in the sediment gets thinner. There also seems to be some confusion with respect to the upper boundary of the GHSZ and the upper boundary of gas hydrate-bearing sediments. The upper boundary of the GHSZ is (with typical water column temperature and in water depths deeper than about 300 m) found in the water column. However, hydrate formation does not occur in the shallow sediments due to a lack of methane, which is lost to the SMT overlying the gas hydrate-bearing sediments. Methane hydrates also constantly dissolve and release gas from the upper hydrate layers due to the concentration gradient produced by AOM occuring in the overlying SMT. This occurs even if hydrates are well within the hydrate stability zone due to undersaturation of the surrounding pore water with respect to methane (cf. Lapham et al., 2010, EPSL; Kasten et al., 2012, Geo-Marine Lett.).

**We agree that the thermodynamic stability domain of gas hydrates is defined by temperature and pressure and is not the same as the zone where gas hydrates are found in the sediment. Methane concentrations must be at saturation for hydrates to form in the thermodynamic stability domain. As Dr. Kasten points out, the upper thermodynamic limit is in the water column in environments where the sediment is several hundred meters below the surface, whereas the actual methane hydrate limit is in the sediment, where methane is present in sufficient quantities. As sea level drops, the upper limit of thermodynamic stability becomes deeper in the water column, but does not change the actual upper limit in the sediment. Our scenario involves the change in depth of the lower limit of stability as sea level drops and the GHSZ thins (or the lower thermodynamic limit moves up, as Dr. Kasten points out), which generates an upward flow of methane and raises the actual upper limit of methane hydrate stability. In the original version of the manuscript we wrote "With the initiation of a glacial cycle, the global temperature decreases, ice builds up on the continents, the sea-level falls, the pressure on the seafloor decreases, and the upper and lower boundaries of the methane hydrate stability field in the sediment column shift upward. " In the revised version (lines 367-369), we write "With the initiation of a glacial cycle, […] and the lower boundary of the methane hydrate stability field in the sediment column shifts upward. "**

1. 355: No, as stated above I am not convinced that this necessarily increases the P flux into the oceanic reservoir.

 **See responses above.**

Figure 1: What precisely do you mean with „pulsed" release of phosphate? This is not clear at all and has also not been discussed in the text. Please indicate where and how the two most important particulate carrier phases of P – i.e. OM and Fe-bound P – are transported into the reservoirs.

**We use pulsed as opposed to steady or invariable.**

Figure 2 needs a complete overhaul. The caption of the figure does not correspond to what is shown in the figure (e.g., the profiles of methane and Fe2+ are not shown) and for part of the profiles it is not clear what is shown (what are Fe oxides and what are Fe sulfide minerals?).

Also the schematic representation in this figure does not correspond or represent what the authors discuss. The phosphate profiles shown (seems that they have been adopted from gravity core data of Niewöhner et al. (1998)) have uniform concentrations in the uppermost part of the sedimentary column. This means that there is definitely no diffusive flux of P across the sediment/water interface – neither during interglacials nor glacials.

**As stated above, Figure 2 was revised, including the addition of scales, and the caption expanded to explain the presence of a nearly vertical SRP concentration immediately below the SWI, as they are typically buffered by absorption to detrital and authigenic iron oxides. Pore-water Fe(II) concentrations are typically very low in the oxic sediment (< 2μM), increase slightly with depth in response to dissimilatory iron reduction, but return to very low concentrations in the sulfate reduction zone as iron sulfides, AVS and pyrite, are precipitated.**

Table 1: Point 6, column on the right: no, as already outlined above it is definitely not true for continental slope/margin sediments that the most reactive forms of Fe occur in the upper part of the sediment. Please revise and specify.

**The kinetics of iron oxide reduction by H₂S depends on the reactivity of the oxides, itself a function of the mineralogy, time since deposition, …**

Point 10, left column: it has to be „increases" instead of „lowers"

**We agree, we substituted "lowers" for "increases" in the revised manuscript.**

---

## Author Response (AR2)

**RESPONSE TO THE REVIEWER'S COMMENTS**
**(Note that our responses to the comments are in boldface.)**

**We thank the associate editor and reviewer for their incisive comments as well as for the opportunity to address these. There is no doubt that the revised manuscript has benefitted from their comments.**

Reviewer #3 expressed dissatisfaction with your revision of the manuscript, because despite providing sufficient responses in the discussion, insufficient implementation into the manuscript was provided.

**All of the reviewer's comments were addressed in detail in our responses and appropriate modifications to the manuscript were implemented. The implementations alerted the reader to the concerns raised by this reviewer, but we did not expand extensively on these within the revised manuscript as we did not want to unduly lengthen the manuscript.**

Aside from revisiting the original suggestions made by Reviewer #3 (and implementing them as applicable), Reviewer #3 suggests a stronger and more accurate literature discussion to compare methane seep sites featuring different methane fluxes/SMTs with the respective phosphate profiles/fluxes.

**We are afraid that the reviewer is missing the point and has not read our reply carefully. The methane flux and the position of the SMT do not alter the phosphate profiles or its flux at the sediment-water interface. Only an upward, vertical migration of the SMT, induced by a destabilization of methane hydrates or an increased flux, will trigger the dissolution of phosphate-laden iron oxides that accumulated above the SMT prior to its vertical migration. Only then will the pore-water phosphate concentration gradient above the SMT increase and potentially increase the flux of phosphate across the sediment-water interface. With time, as the phosphate-laden iron oxides are exhausted, the phosphate gradient will decrease and, thus, its upward flux will subside.**

The reviewer is making the important point (based on field measurements and published literature) that high methane fluxes do not necessarily have to be correlated with high phosphate fluxes. Some examples show the opposite trend caused by geochemical features currently not considered in your manuscript. It therefore seems oversimplified that sea-level fall (and associated increase in methane fluxes) would automatically lead to an increase in phosphate fluxes.

**We absolutely agree with this point. In fact, if the methane flux has been persistent and/or methane seeps out of the sediment (or if the SMT is close to the sediment-water interface), we would not expect a significant phosphate flux out of the sediment, because the phosphate-bearing oxides would either not accumulate in these sediments or have long been dissolved by the sulfidic pore waters. Also see our response to the previous comment above. Furthermore, as indicated in the original and revised manuscript, the phosphate flux will be episodic. In other words, an increase in the pore-water phosphate gradient and a phosphate flux across the sediment-water interface should follow an upward, vertical migration of the SMT in response to a lowering of the sea level, but both the gradient and the flux will wane with time as the phosphate-laden iron oxides are reduced by sulfide produced by AMO within the vertical interval of the SMT**

**migration and are ultimately exhausted. There will be no increase in phosphate gradient and, consequently, no increase in phosphate flux across the sediment-water interface if the amount of phosphate-bearing iron oxides within this interval is negligible, as they may not have accumulated within this interval or been reduced in a previous episode of SMT migration. This scenario was not described explicitly in the previous versions of our manuscript but is now presented on lines 270-282 of the annotated, revised manuscript.**

I agree that the mechanism you are proposing is very interesting; but before I can accept your manuscript for publication I like to see a stronger discussion about potential alternative scenarios under certain geochemical settings that can be reasonable assumed based on published data sets.

**We hope that additional precisions about the proposed scenario, as presented above and the revised manuscript (lines 270-282 of the annotated, revised manuscript), will satisfy the editor and Reviewer#3, as these additions provide ample justification for current observations.**

Please follow the (former and current) suggestions by Reviewer #3 and submit a revised version together with:
- track-changes manuscript
- point-by-point response including line numbers in the revised manuscript, in which the reviewer's comments have been addressed.

Please let me know in case you have any questions.
Tina Treude

Dear Dr. Sundby, dear co-authors,

first of all, I would like to apologize for my delayed response. I thank you very much for adressing the points and questions raised in my review of the initial submission in your cover letter. I now also had a detailed look at the revised version. I have to admit that I was a bit disappointed to see that many of the issues that I have brought forward have not been mentioned or clarified in this new version. So, as a consequence many of my initial points/comments still hold and I will not repeated them in detail here.

**As noted above, all comments were addressed in detail in our previous responses and appropriate modifications to the manuscript were implemented. The implementations alerted the reader to the concerns raised by this reviewer, but we did not expand extensively on these within the revised manuscript as we did not want to unduly lengthen the manuscript.**

I would only like to point out again that I think it is very important to more specifically and critically outine in your manuscript that 1) the flux of phosphate across the SWI is ultimately controlled by the concentration gradient across the SWI (and not necessarily by that in the deeper subsurface sediments and/or the depth location of the SMT/steepness of P gradient around the SMT), and that 2) there are almost no data available for pore-water phosphate concentrations in cold-seep surface sediments. This is why it may be very promising to refer to sites from more or less the same study area that display different depths of the SMT and

the corresponding phosphate profiles in the surface sediments proper as also sggested by the editor Dr. Treude. My feeling is that this would significantly strengthen your manuscript.

**We agree with the reviewer, there cannot be a flux of phosphate out of the sediment without a concentration gradient across the sediment-water interface (SWI). In our proposed scenario, a sea-level drop will induce a destabilization of methane hydrates, increase the flux of methane towards the SWI and displace the SMT upwards. The sulfide produced by AMO will reduce, if present, the phosphate-bearing iron oxides within the displacement interval, releasing phosphate and increasing the phosphate concentration gradient between the SMT and SWI, thus supporting an increasing flux of phosphate across the SWI. As indicated above, if phosphate-bearing iron oxides are absent within the displacement interval (either they never accumulated or were reduced by a previous migration of the SMT) then the phosphate concentration gradient would not be altered significantly, as would the flux of phosphate across the SWI. Even if phosphate-bearing iron oxides are present within the displacement interval, the phosphate gradient and flux across the SWI will wane with time as the phosphate-bearing iron oxides are deactivated (by FeS coating or Fe(II) adsorption) or exhausted (reduced). Hence, to re-iterate what is said above, if the methane flux has been persistent and/or methane seeps out of the sediment (or if the SMT is close to the sediment-water interface), we would not expect a significant phosphate flux out of the sediment, because the phosphate-bearing oxides would either not accumulate in these sediments or have long been dissolved by the sulfidic pore waters, as observed by Niewöhner et al. (1998) in sediments of the upwelling area off Namibia. In fact, the pore-water profiles reported by Wunder et al. (2021) in the Church Trough sediments of South Georgia are exactly what we would expect. Hence, one would not expect to find current examples of phosphate fluxes associated with cold methane seeps. A phosphate flux might have accompanied the methane flux when the latter was first initiated, but would have waned with time. These arguments are now presented on lines 270-282 of the annotated, revised manuscript.**

With respect to comment 1): there is generally considerable difference between the shapes of pore-water profiles and thus gradients/fluxes of PO4 between deeper sediments (retrieved by gravity corers) and those of the surface sediments proper. In this context I would like to bring to your attention a recent paper on biogeochemical processes in sediments of South Georgia. In the study by Wunder et al. (2021, The ISME Journal) we have examined surface sediments from 4 sites in the coastal/shelf environment of this archipelago, which are all located close to methane seep sites and thus are affected by active methane seepage. I.e. they exactly represent the kind of nonsteady-state/transient depositional/geochemical environments that you hypothesize to evolve during glacial sealevel-lowstands. As shown in Fig. 2 of this paper there is no correlation at all between the depth of the SMT/the intensity of upward methane flux and the pore-water profile shapes and gradiens of phosphate. In contrast, the site with the highest methane flux and shallowest depth location of the SMT (Church Trough) is the site with the lowest flux of phosphate into the overlying bottom water. What we also see in the solid-phase data of sites in the area is that even in sulfidic sediments the amount of reactive Fe minerals is similar to that at Fe-rich sites – mostly due to the fact that the inner/bulk Fe oxide mineral in shielded from further reduction by the formation of Fe-sulfide coatings around the mineral particles. The study thus also highlights the importance to differentiate between Fe-rich and sulfde-rich sediments continental margin sediments (see below).

**We thank the reviewer for bringing this publication to our attention. As detailed above, the pore-water profiles reported by Wunder et al. (2021) in the Church Trough sediments of South Georgia are exactly what we would expect after the SMT has settled for some time close to the SWI and no phosphate flux across the SWI would be observed. We also agree that the reactive iron minerals can be shielded from further reaction by Fe-sulfide precipitation or even Fe(II) adsorption. The latter deactivation mechanism has been called upon before as a means to inhibit reactive iron mineral reduction, including in the sapropelic sediments of ferriginous Lake Matano (Crowe et al., 2008). See additions at lines 270-281 of the revised, annotated manuscript.**

**Crowe, S.A., Jones C., Katsev S., Magen C., O'Neill A.H., Sturm A., Canfield D., Haffner G.D., Mucci A., Sundby B. and Fowle D.A. (2008) Anoxygenic phototrophs thrive in an Archean Ocean analogue. Proc. National Academy of Sciences 105(41): 15938-15943.**

Without any doubt there is release of phosphate into the overlying water column across the SWI (e.g. see global map of benthic phosphate fluxes by Hensen et al. ) due to diffusion and additionally by activity of benthic fauna. Nowhere in my comments did I put this in question. I also fully agree – and there is in fact quite some evidence – that phosphate can diffuse upward and downward over considerable distances in the seabed. The depth or source/process from/by which phosphate is liberated into the pore water can, however, be quite different/differ considerably. There is considerable difference between organic-rich/sulfidic sedimentary environments and those rich in iron.
I also fully agree that a decrease in hydrostatic pressure during glacial sealevel lowstands (among other processes) can increase the upward flux of methane and shift the SMT to a shallower sediment depth – thus (of course) also increasing the steepness of the upward directed sulfate gradient and hence the upward diffusive flux of sulfate.

However, based on this you then hypothesize that also the upward flux of phosphate increases from sources deeper in the sediment (namely the SMT) resulting in an increased flux of phosphate across the sediment/water interface. I find it hard to agree with this hypothesis for at least two reasons.

First, in organic-rich sediments (as underlying high-productivity upwelling areas; e.g. Niewöhner et al., 1998) phosphate is not liberated into the pore water at the SMT but diffuses up from much greater sediment depth most likely as a consequence of organic carbon mineralization in deeper sediments (please check again the figures in Niewöhner et al., 1998). As a consequence there is no reason why the upward directed phosphate concentration gradient should change if the SMT moves upward during glacial times.

**We agree with this observation, but as highlighted above, an upward displacement of the SMT, such as would be induced by a sea-level low stand, would burn through a layer (within the displacement interval of the SMT) of phosphate-bearing iron oxides and release phosphate to the pore waters. If the position of the SMT is invariant, one would not expect to see much phosphate released at the SMT, as reported in Niewöhner et al. (1998). See additions at lines 270-282 of the annotated, revised manuscript.**

Second, it could be that the upward directed phosphate flux from the SMT would increase as a consequence of an upward shift of the SMT as is the case in high-accumulation settings where large amounts of reactive Fe minerals are rapidly buried and where phosphate is primarily released at the SMT by reductive dissolution of Fe (III) minerals with hydrogen sulfide (e.g. Riedinger et al., 2005, 2014; März et al., 2008, 2018). This does, however, not

mean that this elevated phosphate flux also „makes it" across the SWI. Please, also compare the paper by Wunder et al. (2021, The ISME Journal).

**It may well be if metal oxide surfaces within the oxic layer of the sediment are not saturated with phosphate, but we found that detrital and diagenetic iron oxides in organic-rich sediments have very high but sliding buffering adsorption capacities. Hence, if the pore-water phosphate flux to the SWI is increased, much of the phosphate will be intercepted by these oxides, but the concentration of maximum buffering capacity of the sediment (or zero equilibrium phosphate concentration (EPC$_0$)) will increase and the concentration gradient and flux of phosphate across the SWI will also increase. Accordingly, we have modified the text and it now reads (lines 291-298 of the annotated, revised manuscript): " Irrespective, detrital and diagenetic iron oxides in organic-rich sediments have very high but sliding buffering adsorption capacities (Sundby et al., 1992). Hence, if the pore-water phosphate flux to the SWI is increased, much of the phosphate will be intercepted by these oxides, but the concentration of maximum buffering capacity of the sediment (or zero equilibrium phosphate concentration (EPC$_0$)) will increase and the concentration gradient and flux of phosphate across the SWI will also increase (Froelich et al., 1988; Sundby et al., 1992). Sorption should therefore not fully restrict the transport of phosphate diffusing up from the SMT towards and across the sediment-water interface.".**

Specific comments

L. 46: methane is also released into the surrounding pore water and diffuses upward if gas hydrates are well within the hydrate stability field (cf. Egorov et al., 1999, Geo-Marine Letters; Lapham et al., 2010, Earth Planet. Sci. Lett.). It is a matter of saturation state of the surrounding pore water with respect to methane and the rate of upward diffusion of methane from the hydrates/hydrate-bearing sediments into the overlying sediments.

**Whereas this information may be factual, it should not be added here since we are, for the sake of setting up the context of our work, paraphrasing a scenario previously proposed by Paull et al. (1991).**

Ls. 87 ff.: phosphate release by sulfide generated by AOM

**The text in parentheses was replaced by "phosphate released to the pore waters upon the reductive dissolution of iron oxides by sulfide produced during anaerobic methane oxidation".**

L 121/122: This statement only holds for high accumulation continental margin settings rich in terrigenous material (like off larger river mouths/fans like the Amazon, Congo, La Plata, Zambesi, etc.). In theses sediments relatively large amounts of reactive Fe(III) are rapidly buried and associated phosphate is primarily released into the pore water by reductive dissolution of Fe (III) minerals with hydrogen sulfide at the SMT (e.g. Riedinger et al., 2005, 2014; März et al. 2008, 2018). These sites often only show a thin sulfidic zone around the SMT.
However, in organic-rich sediments – like those underlying upwelling areas like off Namibia - phosphate is not liberated at the SMT but diffuses up from significantly greater sediment

depth where it is most likely released by deep subsurface organic carbon degradation (cf. Niewöhner et al., 1998).

**Agreed, the text was modified and now reads "According to the depositional setting, pore-water phosphate profiles in marine sediments tend to display a concentration maximum below the dissimilatory iron and sulfate reduction zone (e.g., Krom and Berner, 1981), within or near the sulfate-methane transition zone (e.g., März et al., 2008) as well as at greater depths where it is most likely released by deep subsurface organic carbon degradation (e.g., Niewöhner et al., 1998)."**

L. 140: the total sum

**The "sum total" is the proper English expression and, thus, it was not modified.**

L. 149: bioirrigation „and bioturbation"

**Bioturbation was added to the text of the revised manuscript.**

Ls. 168 ff.:In this paragraph you have not addressed two further fundamental controls on the preservation/burial of both „primary" and authigenic Fe (III) minerals - namely sedimentation/accumulation rate and sulfide exposure time. I would suggest to add these as well.

**The text was modified accordingly.**

L. 178: This statement is still too general. Several more recent studies have shown that this is not necessarily the case …. Please check the papers by Riedinger et al. (2005, 2014, 2017), Egger et al. (2015, 2017) and März et al. (2008, 2018).

**This is what we have observed in estuarine and coastal sediments on both sides of the North Atlantic but, according to the papers the reviewer directs us to, this may not always be the case. Hence, we have tempered the statement. It now reads: " … the bulk reactivity of the sedimentary iron oxides typically decreases with time …".**

L. 250 ff.: This is not necessarily what we see when the SMT moves upward. If the SMT migrates upward (i.e. into sediments which have not yet been exposed to high sulfide concentrations) most of the sulfide produced in the first stage of this transitient situation is immediately trapped in the solid phase (cf e.g. Riedinger et al., 2005).

**We agree that much of the sulfide originating from AMO will likely be trapped by precipitation of a solid sulfide and little will diffuse beyond the SMT. This does not invalidate our proposed scenario of phosphate release to the pore waters as the sulfide will reduce iron oxides before solid sulfides are precipitated, it only restricts the locus of phosphate remobilization to the vicinity of the SMT. In response, we have slightly modified the statement, it now reads: "The upward displacement and ultimate location of the SMT in the sediment column is critical since the reduction of iron oxides by sulfide occurs at this location and within the displacement interval and releases phosphate to the pore water. The closer the SMT migrates to the sediment surface, the greater is the instantaneous flux of sulfate …".**

L. 255 ff.: As outlined in my review of the initial submission it is in no way clear that during periods of a shallower depth SMT (and a steeper gradient and higher upward flux of P from the SMT) also the flux of phosphate across the sediment/water interface is increased. Cf. Wunder et al. (2021).

**Again, as emphasized above, an increased phosphate flux is only expected soon after an upward vertical migration of the SMT if phosphate-bearing iron oxides are present within the displacement interval. Once these oxides are deactivated or exhausted, both the concentration gradient and flux of phosphate will wane. This detail did not appear in the original and first revision, but was added to the latest revision (Lines 270-282 of the annotated, revised manuscript).**

L. 269: … can form carbonate fluorapatite and vivianite …..

**Agreed, vivianite was added to the text of the revised manuscript.**

Ls. 277 and 326: has to be Niew"ö"hner

**Corrected as requested.**

L. 319: But this needs to be shown.

**As it cannot readily be demonstrated, we mitigated the statement by adding "potentially" to the sentence.**

L. 331: yes, I agree …. It could increase the flux …. But not necessarily …

**We have added several caveats to our proposed scenario throughout the revised manuscript and Table 1, these will hopefully satisfy the editor and this reviewer.**

**It is interesting to note that, according to ResearchGate, the preprint of this paper (Biogeosciences Discussions) has already been read more than 106 times and recommended by two readers. After all, the original manuscript received two very laudable reviews from two reputable experts in the field. Hence, we are grateful for the opportunity to share the results of our deliberations.**